# KERNELBAND: Steering LLM-Based Kernel Optimization via Hardware-Aware Multi-Armed Bandits

**Dezhi Ran** [* 1 2]   **Shuxiao Xie** [* 2 3]   **Mingfang Ji** [2 4]   **Anmin Liu** [1]   **Mengzhou Wu** [1 2]   **Yuan Cao** [1 2]   **Yuzhe Guo** [1 2]
**Hao Yu** [5]   **Linyi Li** [6]   **Yitao Hu** [4]   **Wei Yang** [7]   **Tao Xie** [1 2 8 9]

## Abstract

High-performance GPU kernels are critical for efficient Large Language Model (LLM) serving, and yet their optimization remains a bottleneck requiring deep system expertise. Although code LLMs show promise in generating functionally correct code, kernel optimization is intrinsically a search problem over a vast optimization space. This fundamental mismatch prevents existing LLM agents from efficiently exploring the optimization space for diverse hardware and compute patterns. To bridge the gap, we present KERNELBAND, a framework that formulates kernel optimization as a Multi-Armed Bandit (MAB) problem, explicitly balancing exploration and exploitation to guide code LLMs. To navigate the infinite arm space of optimization strategies applied to candidate kernels, we design two key mechanisms: a hardware-aware pruning strategy via profiling bounds and a runtime-behavior clustering algorithm that leverages Lipschitz continuity. Extensive experiments on TritonBench-G with three GPU architectures and four code LLMs show that KERNELBAND consistently outperforms the strongest available agent baseline, achieving up to **1.91×** geometric mean speedup over correctly optimized kernels with **39–140%** relative improvement in Fast@1 success rate. Our

code is available at `https://github.com/TongmingLAIC/KernelBand`.

## 1. Introduction

The computational demands of Large Language Models (LLMs) have grown rapidly (Zhao et al., 2023; Naveed et al., 2025; Floridi & Chiriatti, 2020; Team et al., 2025; 2023; Bai et al., 2023), making efficient serving infrastructure a critical priority (Miao et al., 2025; Ye et al., 2025; Kwon et al., 2023; Fang et al., 2021; Yan et al., 2018; Lin et al., 2024; Pan et al., 2024). At the heart of the efficiency lies kernel optimization (Filipovič et al., 2015; Ryoo et al., 2008; Lange et al., 2025), the engineering of high-performance primitives for fundamental operations such as attention (Dong et al., 2024) and general matrix multiplication (GEMM) (Faingnaert et al., 2022). Traditionally, developing these kernels has been the exclusive domain of specialized human experts, requiring careful manual mapping of algorithms to complex hardware features, such as multi-level memory hierarchies and tensor core instructions (NVIDIA, 2025). To democratize this process, domain-specific languages (DSLs) such as Triton (Tillet et al., 2019) and TileLang (Wang et al., 2025b) have emerged, offering abstractions that hide low-level intricacies. However, although DSLs simplify implementation, achieving peak performance still relies on compiler autotuning mechanisms to select configuration parameters (Gao et al., 2025; Zhu et al., 2022; Wang et al., 2024). As hardware complexity increases, the configuration space suffers from a combinatorial explosion (Gao et al., 2025; Ansel et al., 2024; Shi et al., 2023), rendering heuristic strategies computationally prohibitive and often incapable of finding global optima within reasonable timeframes.

To circumvent the combinatorial explosion, the community has recently turned to code LLMs (Dong et al., 2025; Guo et al., 2024; Hui et al., 2024; Achiam et al., 2023; Caruccio et al., 2024), leveraging their generative capabilities to automate kernel optimization. Existing methods primarily fall into two categories: agent-based methods (Dong et al., 2026; Wang et al., 2025a; Zhang et al., 2025), which use iterative feedback loops to refine implementations, and training-based methods (Baronio et al., 2025; Woo et al., 2025;

---

[*]Equal contribution  [1]Key Lab of HCST (PKU), MOE; SCS, Peking University, Beijing, China [2]Beijing Tongming Lake Information Technology Application Innovation Center (TLAIC), China [3]East China Normal University, Shanghai, China [4]Department of Computer Science, Tianjin University, Tianjin, China [5]Hong Kong University of Science and Technology, Hong Kong, China [6]School of Computing Science, Simon Fraser University, Burnaby, BC, Canada [7]University of Texas at Dallas, USA [8]Fudan University Institute of Systems for Advanced Computing, China [9]Shanghai Institute of Systems for Open Computing, China. Correspondence to: Tao Xie <taoxie@pku.edu.cn>.

*Proceedings of the 43rd International Conference on Machine Learning*, Seoul, South Korea. PMLR 306, 2026. Copyright 2026 by the author(s).

Kong et al., 2025), which fine-tune models on optimization-specific datasets. However, a fundamental mismatch persists: LLMs are inherently trained to generate statistically probable and functionally correct code, whereas kernel optimization is intrinsically a search problem over a vast and discontinuous optimization space (Gao et al., 2025). Consequently, existing agents struggle to balance exploration and exploitation, frequently converging to suboptimal local minima or wasting computational resources on invalid configurations, thereby failing to achieve expert-level performance.

To bridge the gap, we present KERNELBAND, to our knowledge, the first Multi-Armed Bandit (MAB) (Mahajan & Teneketzis, 2008; Scott, 2010; Boursier & Perchet, 2024; Silva et al., 2022) framework for steering LLM-based Triton kernel optimization with provable efficiency under a static-cluster surrogate, enabled by two novel designs. Distinguished from previous methods (Zhang et al., 2025; Kong et al., 2025) that rely on unguided generation or self-reflection heuristics, KERNELBAND formulates the problem in a contextual bandit setting (Bouneffouf et al., 2020), where each arm corresponds to the application of an optimization strategy (Ryoo et al., 2008) (e.g., tiling, vectorization) to a candidate kernel implementation; for tractability, KERNELBAND groups similar kernels into clusters and maintains arms at the cluster level. To address the challenge of infinite action spaces, we design hardware-aware pruning, which uses profiling data to establish tight reward upper bounds, and runtime-behavior clustering, which exploits the Lipschitz continuity (Hager, 1979) of runtime behaviors to estimate rewards for unexplored arms. Theoretically, we prove that KERNELBAND reduces the regret bound to depend on the compact covering number of runtime clusters rather than the vast kernel space.

We evaluate KERNELBAND on TritonBench-G (Li et al., 2025b) across three graphics processing unit (GPU) architectures (RTX 4090, H20, and A100) and four frontier code LLMs (DeepSeek AI, 2025; OpenAI, 2025; Anthropic, 2025; Google, 2025). Empirical results demonstrate consistent superiority over state-of-the-art baselines, achieving up to $1.91\times$ geometric mean speedup and improving the Fast@1 success rate by $39$–$140\%$. Ablation studies confirm that structured exploration is foundational: replacing our bandit policy with LLM-based semantic reasoning regresses performance to $0.97\times$ (below the reference kernel), validating that learned execution statistics outperform intuition. Furthermore, KERNELBAND automatically adapts strategies to hardware bottlenecks and, at a fixed API budget, delivers $35$–$50\%$ higher speedup than unguided methods.

This paper makes the following main contributions:

- We propose KERNELBAND, the first framework to formulate LLM-based kernel optimization as an MAB

problem, effectively resolving the mismatch between functional code generation and optimization search.

- We design a structured acquisition strategy that combines hardware-aware pruning with runtime-behavior clustering, proved to efficiently explore the kernel optimization space under a static-cluster surrogate.

- We demonstrate, through extensive experiments on TritonBench-G across three GPUs and four code LLMs, that KERNELBAND consistently outperforms state-of-the-art methods, achieving up to $1.91\times$ geometric mean speedup, with ablation results confirming the structured bandit policy as the key driver of the gains.

## 2. Problem Formulation

In this section, we formalize kernel optimization as a search problem over a generated code space and subsequently model it as a structured contextual MAB problem with an expanding action space. We explicitly address the mismatch between the generative capabilities of code LLMs (which prioritize functional correctness) and the navigational requirements of kernel optimization (which is intrinsically a search problem over performance landscapes).

### 2.1. The Search Problem

Given a target hardware platform $\mathcal{H}$ and a naive kernel implementation, we seek a kernel $k^*$ minimizing execution time $\mathcal{T}(k, \mathcal{H})$ while preserving functional correctness:

$$k^* = \operatorname*{argmin}_{k \in \mathcal{K}_{\text{valid}}} \mathcal{T}(k, \mathcal{H}) \tag{1}$$

where $\mathcal{K}_{\text{valid}}$ contains all correct implementations.

We view optimization as traversing a directed graph $\mathcal{G} = (\mathcal{V}, \mathcal{E})$: nodes $\mathcal{V}$ are valid kernels; edges $(k \to k') \in \mathcal{E}$ represent applying a strategy $s \in \mathcal{S}$ via a code LLM.

**The fundamental mismatch.** Although modern code LLMs excel at generating functionally correct nodes (kernels), they lack the hardware-specific intuition needed to select edges that efficiently navigate toward performance-optimal regions of $\mathcal{G}$. A naive LLM-based optimizer performs what amounts to a random walk on the graph, wasting substantial effort on transformations that yield negligible or negative speedups. Our goal is to replace this undirected exploration with a principled decision policy that leverages both program semantics and hardware behavior, as illustrated in Figure 1.

### 2.2. Contextual Bandit Formulation with Expanding Action Space

We frame kernel optimization as an **iterative candidate-pool expansion** process. Starting from an initial implemen-

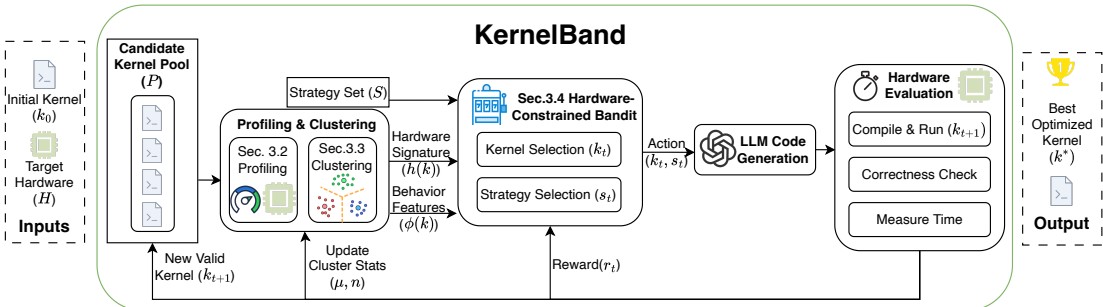

*Figure 1.* Overview of KERNELBAND. Given an initial kernel and the target hardware, KERNELBAND profiles and clusters the candidate kernels, then a hardware-constrained bandit selects a kernel and an optimization strategy; an LLM applies the strategy, and the resulting kernel is compiled, checked for correctness, and timed, with the measured reward updating the bandit. KERNELBAND returns the fastest correct kernel.

tation $\mathcal{P}_0 = \{k_{\text{naive}}\}$, at each step $t$, the agent maintains a candidate pool $\mathcal{P}_t \subseteq \mathcal{V}$ of promising kernels discovered so far. The core decision is to determine which kernel $k \in \mathcal{P}_t$ to expand and which optimization strategy $s \in \mathcal{S}$ to apply.

We model this process as a **contextual bandit problem** rather than full Reinforcement Learning (RL) for three key reasons: first, optimization trajectories are short, making immediate reward predictive of final performance; second, variance in LLM generations complicates long-horizon value estimation; and third, bandits offer rigorous regret bounds under our structural assumptions, ensuring sample efficiency.

**Formal model.** We formulate the optimization process as a contextual bandit defined by a tuple of context, action, transition, and reward. The **context space** consists of behavior-space vectors $\phi(k) \in \mathbb{R}^d$ for each candidate $k \in \mathcal{P}_t$, encoding dynamic execution traces (instantiated as a 5-dimensional vector in Section 3.2). The **action space** at time $t$, denoted $\mathcal{A}_t = \mathcal{P}_t \times \mathcal{S}$, represents applying strategy $s$ to kernel $k$. Because the candidate pool $|\mathcal{P}_t|$ grows with $t$, $\mathcal{A}_t$ is unbounded, distinguishing this setting from standard fixed-arm bandits. Executing action $a_t = (k_t, s_t)$ triggers a **generative transition** $k'_t \sim P_{\text{LLM}}(\cdot \mid k_t, s_t, \mathcal{H})$, where stochasticity stems from the LLM's sampling process. Finally, the agent observes a **reward signal** $r_t \in [0, 1]$ measuring normalized execution-time improvement: $r_t = \text{Clip}(\frac{\mathcal{T}(k_t) - \mathcal{T}(k'_t)}{\mathcal{T}(k_t)}, 0, 1)$, where zero reward is assigned to performance regressions or compilation failures. The agent's objective is to maximize the cumulative expected reward $\sum_{t=1}^{T} \mathbb{E}[r_t]$.

### 2.3. Structural Properties Enabling Tractable Learning

The expanding action space $\mathcal{A}_t$ poses a fundamental challenge: standard bandit algorithms would suffer linear regret $O(T)$. However, kernel optimization exhibits two key structural properties that make the problem tractable:

**Assumption 1 (Hardware-Aware Gain Boundedness)**
*The expected improvement from applying strategy $s$ to kernel $k$ is bounded by hardware limits. Formally, there exists a known bounding function (Williams et al., 2009) $B : \mathcal{V} \times \mathcal{S} \to [0, 1]$ such that*

$$\mu(k, s) \triangleq \mathbb{E}[r_t|k, s] \leq B(k, s) \tag{2}$$

*where $B(k, s)$ estimates the maximum achievable gain based on the gap between $k$'s current performance and the theoretical limit for strategy $s$ on hardware $\mathcal{H}$.*

**Assumption 2 (Lipschitz Continuity in Behavior Space)**
*The expected reward function $\mu(k, s)$ is Lipschitz continuous with respect to execution behavior. That is, for any strategy $s$ and kernels $k_1, k_2$,*

$$|\mu(k_1, s) - \mu(k_2, s)| \leq L_{Lip} \cdot \|\phi(k_1) - \phi(k_2)\|_2 \tag{3}$$

*where $\phi(k)$ captures runtime characteristics. This continuity implies that kernels with similar bottlenecks respond similarly to optimizations; we empirically validate this regularity in Section 4.3.2.*

**Intuition.** Together, these assumptions imply that although the code space is infinite, the space of *meaningful optimization decisions* is low-dimensional and structured, motivating the design of KERNELBAND.

## 3. Methodology: KERNELBAND

### 3.1. System Overview

As shown in Figure 1, KERNELBAND addresses the challenge of the expanding action space $\mathcal{A}_t = \mathcal{P}_t \times \mathcal{S}$ through three key components: (1) **runtime behavior characterization** (Section 3.2), which extracts execution signatures to enable knowledge sharing; (2) **runtime-behavior clustering** (Section 3.3), which groups similar kernels to manage the expanding action space; and (3) **hardware-constrained bandit policy** (Section 3.4), which selects promising optimizations while pruning physically invalid strategies.

*Table 1.* Mapping from optimization strategies to the targeted dimension of $h(k)$. A strategy is pruned for cluster $C_i$ when its target dimension on $\hat{h}_c^{(i)}$ exceeds the saturation threshold $\theta_{\text{sat}}=75\%$.

| Strategy | Target dim. | Primary mechanism |
|---|---|---|
| Tiling | SM | Compute parallelism |
| Vectorization | DRAM | Wider memory transactions |
| Access & Layout | DRAM | Coalesced access patterns |
| Pipeline | DRAM | Compute/memory overlap |
| Fusion | L2 | Reduced intermediate writes |
| Reordering | L2 | Improved cache locality |

## 3.2. Runtime Behavior Characterization

We characterize each kernel $k$ using two complementary representations to satisfy the smoothness and boundedness assumptions defined in Section 2.

**Behavioral feature vector $\phi(k)$.** For clustering kernels with similar optimization responses, we define a 5-dimensional vector $\phi(k)$ comprising normalized execution time $\tilde{\mathcal{T}}(k)$ and hardware counters:

$$\phi(k) = [\tilde{\mathcal{T}}(k), \ n_{\text{reg}}, \ n_{\text{smem}}, \ d_{\text{block}}, \ \eta_{\text{occ}}] \qquad (4)$$

representing registers per thread, shared memory per block, block dimension, and occupancy, respectively. Kernels close in $\phi$-space share similar bottlenecks (Assumption 2), allowing the bandit to generalize strategy performance.

**Hardware signature $h(k)$.** To implement the bounding function $B(k,s)$ from Assumption 1, we extract a hardware signature $h(k)$ using NVIDIA Nsight Compute (NCU) (NVIDIA Corporation, 2024), measuring peak throughput percentages for DRAM, L2 cache, and the streaming multiprocessor (SM). These metrics identify the dominant bottleneck (memory bandwidth/compute/cache) for pruning physically implausible optimization strategies; Table 1 summarizes the strategy-to-target mapping that drives the mask in Eq. (6).

## 3.3. Structured Exploration via Dynamic Clustering

**Periodic re-clustering.** Instead of maintaining a separate bandit arm for every kernel, we maintain arms for kernel *clusters*. At iteration $t$, we partition the candidate pool $\mathcal{P}_t$ into $K$ clusters $\mathcal{C}_t = \{C_1, \ldots, C_K\}$ using K-Means on $\{\phi(k)\}$. Clusters are recomputed every $\tau$ iterations. This periodic update balances the need to track shifting kernel behaviors with the stability required for bandit convergence.

**Representative profiling.** Because hardware profiling is expensive ($\approx 10$s), we profile only the centroid kernel $k_c^{(i)}$ of each active cluster during the re-clustering phase. We approximate the hardware constraints of the entire cluster using this representative, substantially reducing overhead.

## 3.4. Hardware-Constrained Bandit Policy

As depicted in Algorithm 1, we formulate the decision process as a *hardware-constrained bandit* whose action selection masks physically invalid strategies.

**Pruning via hardware potential.** We define a binary mask $M_{i,s} \in \{0,1\}$ based on the hardware signature $h(k)$. A strategy $s$ is valid for cluster $C_i$ only if it targets a non-saturated resource:

$$M_{i,s} = \mathbb{I}\left[h(k_c^{(i)})[\text{Target}(s)] < \theta_{\text{sat}}\right] \qquad (5)$$

where $\theta_{\text{sat}}$ is a saturation threshold (e.g., 75%). This strategy reduces the effective action space from $|\mathcal{S}|$ to $|\mathcal{S}_{\text{valid}}|$.

**Action selection.** At iteration $t$, we select a cluster-strategy pair $(I_t, S_t)$ using a Masked Upper Confidence Bound (UCB) index. Let $\hat{\mu}_{i,s}$ be the empirical mean reward and $N_{i,s}$ the visit count. We maximize UCB only among valid actions:

$$(I_t, S_t) = \underset{\substack{i \in [K], s \in \mathcal{S} \\ \text{s.t. } M_{i,s}=1}}{\operatorname{argmax}} \left(\hat{\mu}_{i,s} + c\sqrt{\frac{\ln t}{N_{i,s}}}\right) \qquad (6)$$

Once $(I_t, S_t)$ is chosen, we sample a specific kernel $k_t \in C_{I_t}$ via a softmax distribution over local potential scores $\hat{V}_{\text{hw}}(k, S_t) = \theta_{\text{sat}} - \hat{h}_c^{(I_t)}[\text{Target}(S_t)]$, measuring the remaining headroom for strategy $S_t$ on cluster $C_{I_t}$. Because the centroid signature $\hat{h}_c^{(I_t)}$ is shared by all kernels in the cluster (Section 3.3), within-cluster sampling reduces to uniform draws in expectation; this centroid-only surrogate trades a slight loss in intra-cluster discrimination for roughly $3\times$ profiling-time savings.

## 3.5. Theoretical Analysis

We analyze KERNELBAND through the lens of $\epsilon$-**optimality**, acknowledging that in high-dimensional kernel optimization, finding a solution within a small tolerance of the global optimum is the practical goal.

**Theorem 1 (Convergence to $\epsilon$-Optimal Solution)** *Let $\mu^*$ be the optimal performance and $r_t$ be the reward at step $t$. Under Assumptions 1 and 2, with probability $1 - \delta$, the average regret satisfies*

$$\frac{1}{T}\sum_{t=1}^{T} \mathbb{E}[\mu^* - r_t] \leq C\sqrt{\frac{K|\mathcal{S}_{valid}|\ln T}{T}} + L_{Lip} \cdot \max_i diam(C_i) \qquad (7)$$

*where $C$ is a constant derived from the UCB analysis.*

The proof is in Appendix I.

**Algorithm 1** Workflow of KERNELBAND.

---

**Require:** Kernel $k_0$, Strategies $\mathcal{S}$, Budget $T$, Period $\tau = 10$, Clusters $K$

1: $\mathcal{P} \leftarrow \{k_0\}, \mathcal{C} \leftarrow \{\mathcal{P}\}$      ▷ *Initial single cluster*
2: Initialize $N_{i,s} \leftarrow 1, \hat{\mu}_{i,s} \leftarrow 0.5$ for all $i, s$
3: Initialize Masks $M_{i,s} \leftarrow 1$      ▷ *No pruning initially*
4: **for** $t = 1$ **to** $T$ **do**
5:      Compute $\phi(k)$ for all $k \in \mathcal{P}$
6:                  ▷ *Periodic clustering & profiling*
7:      **if** $t \bmod \tau = 0$ **and** $|\mathcal{P}| \geq 2K$ **then**
8:          $\mathcal{C} \leftarrow \text{KMeans}(\{\phi(k)\}, K)$
9:          Update centroids $k_c^{(i)}$ and profile $h_c^{(i)} \leftarrow \text{NCU}(k_c^{(i)})$
10:      **end if**
11:                  ▷ *Hardware-constrained selection*
12:      **if** centroids are profiled **then**
13:          $M_{i,s} \leftarrow \mathbb{I}[h_c^{(i)}[\text{Target}(s)] < \theta_{\text{sat}}]$
14:      **end if**
15:      Select $(I_t, S_t)$ via Eq. (6) (Masked UCB)
16:      Sample $k_t \in C_{I_t}$ with $P(k) \propto \exp(\hat{V}_{\text{hw}}(k, S_t))$    ▷ *Centroid surrogate (Section 3.4)*
17:                  ▷ *Generation & update*
18:      $k_t' \leftarrow \text{LLM}(k_t, S_t)$
19:      **if** Verify($k_t'$) **then**
20:          $r_t \leftarrow \text{Clip}((\mathcal{T}(k_t) - \mathcal{T}(k_t'))/\mathcal{T}(k_t), 0, 1)$
21:          $\mathcal{P} \leftarrow \mathcal{P} \cup \{k_t'\}$
22:      **else**
23:          $r_t \leftarrow 0$      ▷ *Verification failure or regression*
24:      **end if**
25:      $N_{I_t, S_t} \leftarrow N_{I_t, S_t} + 1$
26:      $\hat{\mu}_{I_t, S_t} \leftarrow \hat{\mu}_{I_t, S_t} + (r_t - \hat{\mu}_{I_t, S_t})/N_{I_t, S_t}$
27: **end for**
     **return** $\arg\min_{k \in \mathcal{P}} \mathcal{T}(k)$

---

### 3.6. Implementation Details

We employ $|\mathcal{S}| = 6$ optimization strategies: *tiling*, *vectorization*, *fusion*, *pipeline*, *reordering*, and *access & layout* (details in Appendix B). Kernel clustering uses scikit-learn's KMeans with $K = 3$ clusters; clusters are recomputed every $\tau = 10$ iterations. For hardware-aware pruning, we use NCU to record throughput metrics (caching results by code hash), and apply a saturation threshold $\theta_{\text{sat}} = 75\%$ to filter strategies that target already-saturated resources. The UCB exploration parameter is $c = 2.0$. These design choices introduce minimal overhead: feature extraction adds less than 1% runtime cost, and clustering incurs negligible cost, scaling as $O(|\mathcal{P}_t|)$ every $\tau$ iterations. The dominant costs remain LLM generation and kernel compilation (Section 4.3.5); sensitivity to the cluster count $K$ and to individual components is analyzed in Sections 4.3.1 and 4.4, whereas the saturation threshold $\theta_{\text{sat}}$ and strategy-set cardinality are fixed from pilot studies (Appendices B.1–B.2).

## 4. Experiments

### 4.1. Experimental Settings

**Benchmark.** We evaluate on a corrected version of TritonBench-G (Li et al., 2025b), a popular benchmark of GPU Triton kernels (details in Appendix F). After excluding `sin_computation` (that admits trivial simplification yielding artificially high speedups), we obtain 183 kernels spanning 13 functional categories (e.g., attention, matrix multiplication, normalization, fused operations) and 5 difficulty levels (L1–L5).

**Hardware platforms.** We conduct experiments on three NVIDIA GPUs: consumer-grade RTX 4090 along with datacenter-grade H20 and A100 (CUDA 12.1, Triton 3.3.0).

**LLM backend.** We use DeepSeek-V3.2 (DeepSeek AI, 2025) as the primary LLM backend, with additional backends (OpenAI, 2025; Anthropic, 2025; Google, 2025) evaluated in Section 4.3.3 (configurations in Appendix C).

**Baselines.** We use (1) **GEAK** (Wang et al., 2025a), an open-source Triton kernel optimization agent using iterative refinement, with minimal adaptation for our hardware and (2) **Best-of-N (BoN)**, which samples $N = T$ independent variants and selects the fastest (isolating iterative effects). Adaptation details are in Appendix F. We also compare against **PyTorch** baselines, eager execution, `torch.compile` with inductor backend, and `torch.compile` with max-autotune, to contextualize optimization gains with standard PyTorch execution (Appendix G.1).

**Optimization budget.** We optimize each kernel for $T = 20$ iterations by default; scaling analysis (Section 4.3.1) extends to $T = 40$.

**Evaluation methodology.** We adopt TritonBench's hierarchical evaluation: each task optimizes one reference kernel over $T$ iterations. Candidates undergo two-stage correctness verification (*Call Accuracy* for runtime errors; *Execution Accuracy* for numerical equivalence via `torch.allclose`), and then are benchmarked across 10+ input shapes. Per-task speedup is the ratio of total runtimes (baseline over optimized) for the best correct candidate. Details are in Appendix E.

**Metrics.** We report three complementary metrics: (1) **Correct (%)**: percentage of tasks yielding at least one valid kernel ($\geq 1$). (2) **Fast@1 (%)**: percentage of tasks where the best kernel achieves speedup $> 1.0\times$ (failed tasks are counted as 0). (3) **Geometric Mean Speedup**: *standard mode* (tables) averages over correct tasks only; *fallback mode* (figures) assigns $1.0\times$ to failures and regressions, reflecting practical deployment.

*Table 2.* Performance on TritonBench-G, stratified by difficulty level (L1: easiest, L5: hardest). Adjacent levels are merged where sample sizes are small (L1: 3, L5: 5 kernels) to ensure statistically meaningful aggregation. C: Correct (%). F: Fast@1 (%). G: Geometric mean speedup (*standard mode*: computed over correct tasks only, including regressions). Best results per configuration are in **bold**.

| Platform | Method | L1–2 | | | L3 | | | L4–5 | | | All | | |
| | | C | F | G | C | F | G | C | F | G | C | F | G |
|---|---|---|---|---|---|---|---|---|---|---|---|---|---|
| RTX 4090 | BoN | 63.6 | 9.1 | 0.86 | 12.1 | 6.1 | 0.89 | 28.6 | 14.3 | 1.09 | 31.1 | 10.0 | 0.96 |
| | GEAK | 68.2 | 36.4 | 1.62 | 21.2 | 18.2 | 1.45 | 80.0 | 40.0 | 1.34 | 55.6 | 31.1 | 1.44 |
| | KERNELBAND | **81.8** | **50.0** | **2.14** | **60.6** | **30.3** | **1.82** | **91.4** | **51.4** | **1.47** | **77.8** | **43.3** | **1.74** |
| H20 | BoN | 65.5 | 34.5 | 1.41 | 36.9 | 23.1 | **1.35** | 7.1 | 1.4 | 0.63 | 29.3 | 15.9 | 0.99 |
| | GEAK | 68.9 | 34.5 | 1.29 | 60.0 | 29.2 | 1.16 | 31.4 | 14.3 | 0.90 | 49.4 | 23.8 | 1.06 |
| | KERNELBAND | **93.1** | **51.7** | **1.70** | **86.2** | **63.1** | **1.35** | **67.1** | **54.3** | **1.46** | **79.3** | **57.3** | **1.45** |
| A100 | BoN | 73.9 | 34.8 | 1.43 | 38.7 | 24.2 | 1.23 | 14.8 | 6.8 | 0.76 | 31.2 | 16.8 | 0.98 |
| | GEAK | 65.2 | 43.5 | 1.09 | 54.8 | 45.2 | 1.18 | 38.6 | 33.0 | 1.54 | 48.0 | 38.7 | 1.34 |
| | KERNELBAND | **95.7** | **82.6** | **2.23** | **91.9** | **77.4** | **2.05** | **67.1** | **42.1** | **1.75** | **79.8** | **60.1** | **1.91** |

## 4.2. Main Results

**Consistent performance dominance.** As shown in Table 2, KERNELBAND achieves the highest performance across all GPU architectures. On A100, KERNELBAND attains 1.91× geometric mean speedup with 79.8% correctness, outperforming the best-performing baseline, GEAK, by 42.5% in speedup and 66.2% in success rate. We also observe consistent advantages on RTX 4090 (1.74× vs. 1.44×) and H20 (1.45× vs. 1.06×).

**Robust adaptation via hardware-aware optimization.** The most effective kernel optimizations are *hardware-dependent*: the optimal schedule/transform set must reflect each platform's compute-memory balance and architectural constraints (Williams et al., 2009; Chen et al., 2018; Zheng et al., 2020). KERNELBAND adapts its optimization choices across devices rather than applying a fixed, hardware-agnostic search policy. For example, compared to H20, KERNELBAND allocates more exploration budget to FUSION on RTX 4090 (18.5% vs. 12.8%), whereas H20 explores TILING more frequently (10.0% vs. 7.6%). Detailed statistics and analysis can be found in Appendix H. Such platform-specific optimization divergence confirms that our hardware-aware pruning (Assumption 1) effectively calibrates exploration to specific bottlenecks without manual tuning. Consequently, although baseline methods struggle with the diversity (GEAK drops from 1.44× on RTX 4090 to 1.06× on H20), KERNELBAND still maintains robust performance (1.45× to 1.91×) across all the platforms.

**Scaling with kernel complexity.** The performance gap widens on challenging kernels. For the 23 **Hard (L4–5)** kernels on A100, KERNELBAND achieves 67.1% correctness and 1.75× speedup, significantly outperforming GEAK (38.6% C, 1.54× G). Although naive sampling (BoN) fails to generate valid kernels for 85% of the tasks, and unstructured search (GEAK) struggles to improve performance, KERNELBAND's structured exploration efficiently identifies

paths that are both *correct* and *efficient* in the vast optimization space.

**Practical optimization yield.** The Fast@1 metric measures the probability of finding an *optimized* kernel (speedup > 1.0×). KERNELBAND achieves consistent Fast@1 rates of 43.3%–60.1% across all the platforms. In contrast, BoN fails to accelerate most tasks (10.0%–16.8%), whereas GEAK shows both lower average performance and higher volatility (23.8%–38.7%). The 39–140% relative improvement of KERNELBAND over GEAK confirms the effectiveness of our structured exploration: **hardware-aware pruning** suppresses low-headroom strategy choices early, and **runtime-behavior clustering** efficiently guides the search toward high-potential regions. The two algorithmic designs ensure that KERNELBAND not only generates correct code but delivers actual speedups for a substantially larger fraction of the optimization budget.

## 4.3. Detailed Analysis

In this section, we conduct a detailed analysis to validate KERNELBAND's algorithmic properties, generalization capabilities, and practical cost-efficiency. Unless otherwise stated, all the experiments use a 50-kernel subset (preserving category and difficulty distribution as detailed in Appendix D) on the H20 platform.

### 4.3.1. SCALABILITY AND CLUSTERING SENSITIVITY

We analyze the sample efficiency (i.e., scaling to more iterations) and hyperparameter sensitivity by running KERNELBAND with $K \in \{1, 2, 3, 5\}$ against the baselines for an extended budget of $T = 40$ iterations, as shown in Figure 2.

**Superior scaling behavior.** Although the baselines saturate early—BoN stagnates at 1.05× and GEAK plateaus at 1.13× after 25 iterations—KERNELBAND ($K$=3) shows continuous improvement, reaching 1.71× at $T$=40. At the

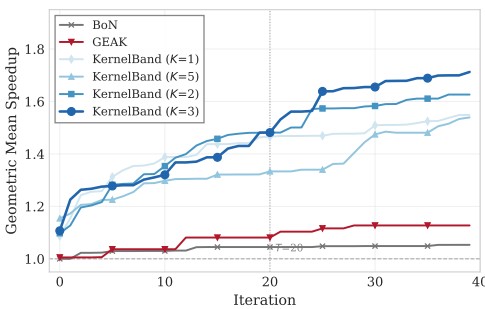

*Figure 2.* Geometric mean speedup vs. iteration budget $T$ for KERNELBAND ($K \in \{1, 2, 3, 5\}$) and the GEAK and BoN baselines (H20 50-kernel subset).

*Table 3.* LLM generalization results. C: Correct (%). F: Fast@1 (%). G: Geometric mean speedup. Best results per model are in **bold**.

| Model | Method | C (%) | F (%) | G |
|---|---|---|---|---|
| DeepSeek-V3.2 | BoN | 27.5 | 12.5 | 1.10 |
| | GEAK | 37.5 | 17.5 | 0.95 |
| | KERNELBAND | **85.0** | **67.5** | **1.52** |
| GPT-5 | BoN | 44.9 | 28.6 | 1.14 |
| | GEAK | 51.0 | 24.5 | 1.07 |
| | KERNELBAND | **81.6** | **65.3** | **1.72** |
| Claude Opus 4.5 | BoN | 40.8 | 24.5 | 1.17 |
| | GEAK | 63.3 | 38.8 | 1.30 |
| | KERNELBAND | **89.8** | **73.5** | **1.82** |
| Gemini 3 Flash | BoN | 47.9 | 25.0 | 1.20 |
| | GEAK | 62.5 | 37.5 | 1.21 |
| | KERNELBAND | **70.8** | **45.8** | **1.48** |

standard budget ($T$=20), KERNELBAND achieves $1.48\times$ speedup, outperforming GEAK by 37%. This result confirms that our MAB solution enables provably efficient exploration (under the static-cluster surrogate analyzed in Section 3.5) for the challenging kernel optimization problem.

**Optimal clustering granularity.** The interplay between cluster count $K$ and iteration budget reveals a clear trade-off. For limited budgets ($T \leq 10$), smaller $K \in \{1, 2\}$ performs best by concentrating exploration. However, as the budget grows ($T \geq 20$), $K = 3$ overtakes simpler configurations (reaching $1.66\times$ vs. $1.58\times$ for $K = 2$ at $T = 30$). The **5% advantage** demonstrates the value of cross-cluster knowledge transfer when sufficient budget is available. $K = 5$ consistently underperforms ($1.54\times$), suggesting that excessive fragmentation hurts bandit learning. We therefore recommend $K = 3$ for default optimization. Importantly, all the tested $K$ configurations consistently outperform both the baselines across the full iteration range, indicating that KERNELBAND's advantage is robust to this hyperparameter.

**Choosing $K$ in practice.** The budget interaction gives a simple rule of thumb: pick $K$=2 for tight budgets ($T < 10$) and $K$=3 otherwise; do not raise $K$ further unless $T$ exceeds 50, because per-arm sample counts degrade faster than what cross-cluster knowledge transfer can compensate. The cardinality of the strategy set $\mathcal{S}$ should track $K$ similarly—we examine this dependence in Appendix B.2.

### 4.3.2. EMPIRICAL VALIDATION OF LIPSCHITZ CONTINUITY

Assumption 2 posits that kernels close in behavior space respond similarly to the same strategy. We empirically validate this assumption on the H20 50-kernel subset with $T$=20 and DeepSeek-V3.2. For each strategy $s$ in the bandit trace, we pair all $(k, s, r)$ triples and record the Euclidean distance $\|\phi(k_i) - \phi(k_j)\|_2$ alongside the absolute reward gap $|r_i - r_j|$. Binning by $\phi$-distance (full table in Ap-

pendix J.1), we find that the mean reward discrepancy rises monotonically with distance; the pairwise Spearman correlation is $\rho$=0.64, and the closest bin shows a $4.5\times$ smaller reward gap than the most distant. This relationship is much tighter than what global Lipschitz continuity would require, yet exactly the local regularity that our cluster-level UCB exploits—the *w/o Clustering* ablation in Table 5 corroborates the practical value of this structure. The $\phi$ representation itself is justified by a leave-one-out feature-importance ablation in Appendix J.3, where no single dimension proves redundant.

### 4.3.3. ROBUSTNESS ACROSS LLM BACKENDS

To ensure that our gains are not model-specific, we benchmark KERNELBAND across four state-of-the-art LLMs.

**Generalized performance advantage.** As shown in Table 3, KERNELBAND consistently outperforms the baselines regardless of the underlying model. With Claude Opus 4.5, it achieves $1.82\times$ speedup and 89.8% correctness. Even with the smaller Gemini 3 Flash, KERNELBAND maintains a substantial lead ($1.48\times$ vs. $1.21\times$ for GEAK).

**Compensating for model capabilities.** Although absolute performance naturally correlates with model strength (Claude > GPT-5 > DeepSeek > Gemini), the *relative* gain provided by KERNELBAND remains robust, indicating that our structured exploration framework acts as an amplifier of any code LLM.

### 4.3.4. STRATEGY SELECTION PATTERN

**Strategy risk-reward profiles.** Table 4 reveals how the bandit policy manages trade-offs between risks and rewards. **Tiling** represents a "high-risk, high-reward" strategy: low success rate (14.4%) but substantial impact when success-

*Table 4.* Strategy selection statistics. Freq: selection frequency (%). Succ: success rate (%, correct & speedup $> 1.0\times$). Best: percentage of successful applications that contribute to the final best kernel.

| Strategy | Freq (%) | Succ (%) | Best (%) |
|---|---|---|---|
| Tiling | 10.0 | 14.4 | 61.5 |
| Vectorization | 14.7 | 57.1 | 17.1 |
| Fusion | 12.8 | 75.0 | 55.2 |
| Pipeline | 9.8 | 64.4 | 26.3 |
| Reordering | 33.2 | 48.7 | 25.3 |
| Access & Layout | 19.5 | 29.5 | 19.2 |

ful (61.5% best-kernel contribution). Conversely, **Vectorization** offers "low-risk, low-reward" gains (57.1% success, 17.1% best). **Fusion** strikes a balance (75.0% success, 55.2% best). The bandit policy of KERNELBAND effectively navigates these profiles, prioritizing reliable gains while occasionally gambling on high-variance strategies.

### 4.3.5. COST AND EFFICIENCY ANALYSIS

Finally, we examine the computational and economic feasibility of KERNELBAND for real-world use.

**Time breakdown.** Figure 3 decomposes the per-kernel iteration latency. Although LLM inference dominates serial execution (87%), batched inference shifts the bottleneck to compilation (34%) and execution (30%), with the remaining $\sim$36% spread across profiling, correctness verification, and bandit/clustering overhead. The effective wall-clock time per iteration drops to 129s, confirming that KERNELBAND remains within practical compilation timeouts.

**Cost analysis.** Despite higher per-iteration costs due to multi-strategy exploration, KERNELBAND delivers superior optimization per dollar as shown in Figure 4. At a fixed budget of $0.50 per kernel, it achieves $1.83\times$ speedup, outperforming GEAK by 35% ($1.35\times$) and BoN by 50% ($1.22\times$). This cost-effectiveness stems from our pruning mechanism, filtering low-value candidates early, ensuring that API costs translate directly to performance gains.

### 4.4. Ablation Study

We dissect the effectiveness of KERNELBAND through single-component ablations (removing one module) and framework-level changes (altering the optimization paradigm). Detailed descriptions and analysis of ablation settings can be found in Appendix K.

**Structured bandit policy is foundational.** As shown in Table 5, the most critical finding is that replacing bandit-based selection with LLM semantic reasoning (*LLM Strategy Selection*, where an LLM selects which strategy to apply

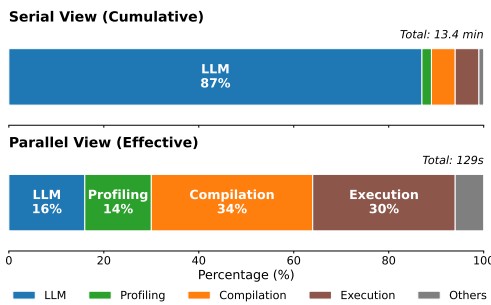

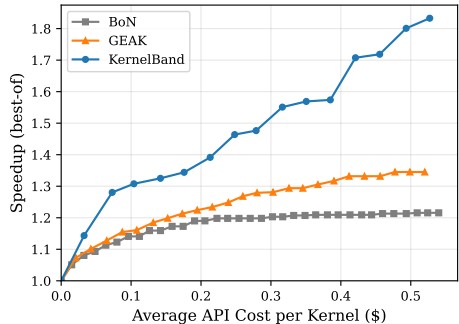

*Figure 3.* Time breakdown per kernel/iteration: (a) serial cumulative time (13.4 min); (b) parallel wall-clock time with batched LLM calls (129s).

*Figure 4.* Geometric mean speedup vs. API cost per kernel (USD) for KERNELBAND and the GEAK and BoN baselines (H20 50-kernel subset).

based on kernel analysis rather than using a bandit policy) causes a sharp regression to $0.97\times$ speedup. This result confirms that learned bandit policies are superior to LLM intuition for strategy selection. Furthermore, removing the strategy set entirely (*w/o Strategy*, where free-form iterative generation is used as GEAK without structured strategies or profiling guidance) drops performance to $1.15\times$. Injecting raw profiling metrics without strategies (*+ Raw Profiling*) further degrades correctness to 43.9%. These results validate that our structured bandit policy is the essential bridge between hardware information and code generation.

**Component contribution hierarchy.** As shown in Table 5, at the standard budget ($T = 20$), profiling guidance proves more critical than clustering: disabling profiling (*w/o Profiling*) causes a 20% speedup drop ($1.57\times \rightarrow 1.26\times$), whereas disabling clustering (*w/o Clustering*) yields a 10% drop ($1.41\times$). However, as shown in Section 4.3.1, clustering's value grows with iteration budget. Both components contribute incrementally to the $2.60\times$ improvement over the BoN baseline ($0.60\times$).

**Structure vs. learned exploration.** The *Masked Random* row cleanly decomposes the contribution of action-space structuring from learned exploration-exploitation. Masked

*Table 5.* Ablation results: single-component ablations (top) and framework-level ablations (bottom) on the H20 50-kernel subset ($T{=}20$).

| Type | Configuration | C (%) | F (%) | G |
|---|---|---|---|---|
| Single | KernelBand (Full) | **87.8** | **63.4** | **1.57** |
| | w/o Clustering ($K{=}1$) | 82.9 | 58.5 | 1.41 |
| | w/o Profiling | 85.4 | 56.1 | 1.26 |
| | Masked Random | 81.3 | 49.2 | 1.23 |
| | LLM Strategy Selection | 68.3 | 36.6 | 0.97 |
| Frame. | w/o Strategy + Raw Prof. | 43.9 | 26.8 | 1.12 |
| | w/o Strategy Set | 78.0 | 48.8 | 1.15 |
| | BoN (baseline) | 34.2 | 17.1 | 0.60 |

Random keeps the strategy set, hardware mask, and cluster partition but draws each (cluster, strategy, kernel) uniformly at random instead of via UCB. Masked Random reaches $1.23\times$—well above BoN ($0.60\times$) but 28% below the full system ($1.57\times$). Hence, structuring the action space already delivers most of the gain over free-form sampling, whereas learned UCB updates supply the remaining $0.34\times$ headroom.

## 5. Related Work

**LLM-based kernel optimization.** Recent and concurrent work explores LLM-based agentic workflows for automated kernel optimization. STARK (Dong et al., 2026) employs multi-agent collaboration with grounded instruction and strategic search to explore the optimization space. CudaForge (Zhang et al., 2025) uses a Coder-Judge architecture that iteratively generates and refines CUDA kernels with profiling feedback from Nsight Compute. GEAK (Wang et al., 2025a) adapts Reflexion-style reasoning loops with inference-time compute scaling for Triton kernel generation. TritonForge (Li et al., 2025a) integrates runtime profiling with iterative code transformation to identify bottlenecks and propose targeted modifications. These methods demonstrate the value of code LLMs but lack principled exploration-exploitation mechanisms to effectively navigate the vast optimization space of kernels.

An alternative paradigm focuses on fine-tuning LLMs specifically for kernel generation. ConCuR (Kong et al., 2025) generates and curates CUDA kernels with reasoning traces for supervised fine-tuning. Kevin (Baronio et al., 2025) applies multi-turn reinforcement learning to address challenges in long-trajectory kernel optimization. TritonRL (Woo et al., 2025) combines supervised fine-tuning with RL using hierarchical reward assignment to mitigate reward hacking. Our work complements this line of work and remains orthogonal to it, as KERNELBAND serves as an amplifier of any code LLM, as our experiments demonstrate.

**Multi-armed bandits and their applications.** Multi-armed bandits (Berry et al., 1997; Slivkins et al., 2019; Kuleshov & Precup, 2014; Vermorel & Mohri, 2005) provide a principled framework for sequential decision-making under uncertainty, with applications ranging from clinical trials to online advertising (Lai & Robbins, 1985). Classical algorithms such as Upper Confidence Bound (Auer et al., 2002) and Thompson Sampling (Thompson, 1933) offer optimal regret bounds for stationary environments, whereas recent work has extended these algorithms to contextual (Li et al., 2010; Chu et al., 2011), metric (Kleinberg et al., 2008), clustering (Gentile et al., 2014), and hierarchical settings (Bubeck et al., 2011). In system optimization, bandit algorithms have shown promise for parameter tuning (Snoek et al., 2012; Pacula et al., 2012), compiler optimization (Xu et al., 2017), and resource allocation (Pandey & Venkatesh, 2025). However, applying bandit algorithms to kernel optimization remained largely unexplored before KERNELBAND, despite the natural fit between the exploration-exploitation trade-off and the challenge of navigating vast optimization spaces.

## 6. Conclusion

We have introduced KERNELBAND, a framework that casts kernel optimization for code LLMs as a hardware-constrained contextual bandit. By combining runtime-behavior clustering with hardware-aware pruning, KERNELBAND enables provably efficient navigation of the optimization space under a static-cluster surrogate. Across three GPUs and four code LLMs, it improves geometric mean speedup by up to $1.91\times$ and Fast@1 by 39–140% over the strongest baseline, underscoring the value of structured, hardware-aware exploration for applying LLMs to systems optimization.

## 7. Limitations

We identify three classes of limitations. First, the local Lipschitz regularity that drives our cluster-level UCB can fail in the following regimes: discrete algorithmic rewrites (e.g., reduction $\rightarrow$ parallel scan), occupancy cliffs from small changes in register or shared-memory pressure, and kernels with near-identical $\phi$ but divergent fine-grained behavior (bank conflicts and irregular access). The UCB policy still converges in such cases, though more slowly than what the average bound predicts. Second, the framework relies on the underlying LLM for valid DSL synthesis; for newer or less-documented DSLs, the gap to expert kernels likely widens. Third, Theorem 1 analyzes a static-cluster surrogate, whereas our algorithm re-clusters every $\tau{=}10$ iterations; the partition is empirically stable (Appendix J.2), so the bound applies approximately between re-clusterings—a fully dynamic analysis remains future work.

## Acknowledgments

This work was supported in part by the Fundamental and Interdisciplinary Disciplines Breakthrough Plan of the Ministry of Education of China (No. JYB2025XDXM118), and by the National Natural Science Foundation of China under Grants U25A6023, 92464301, and 623B2006.

## Impact Statement

This paper introduces KERNELBAND, an automated framework that leverages code LLMs to optimize GPU kernels. We anticipate that, by reducing costs and enhancing the throughput of LLM serving, KERNELBAND will offer substantial benefits in two key areas,

(1) Social and economic impact: This framework facilitates the wider deployment of LLM-based systems by lowering operational barriers. Although we remain aware of potential risks associated with broader AI adoption—such as safety, security, and fairness concerns—we posit that cost-efficient infrastructure accelerates social productivity and promotes AI democratization, thereby helping mitigate economic disparities.

(2) Environmental impact: By optimizing the computational efficiency of core kernels used in LLM inference, KERNELBAND directly reduces demand for hardware resources. Consequently, this reduction leads to improved energy efficiency and lower carbon emissions.

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

*Table 6.* Complete optimization strategy set used by KERNELBAND (six strategies; derivation in Section B.1).

| Strategy | Description |
|---|---|
| Tiling | Partition computation into configurable tile sizes for improved cache locality and parallelism |
| Vectorization | Use vector loads/stores (e.g., float4) for improved memory throughput |
| Fusion | Combine multiple operations to reduce intermediate memory traffic |
| Pipeline | Configure software pipelining depth for latency hiding |
| Reordering | Optimize loop order and instruction scheduling for better ILP |
| Access & Layout | Optimize memory access patterns, coalescing, and data layout |

## A. Profiling and Clustering Features

**Behavioral feature vector** $\phi(k)$**.** As defined in Section 3.2, we extract a 5-dimensional vector for clustering kernels with similar optimization responses:

- $\tilde{\mathcal{T}}(k)$: normalized execution time (from timing run, log-transformed)
- $n_{\text{reg}}$: registers per thread (from `cuFuncGetAttribute`)
- $n_{\text{smem}}$: shared memory per block (from `cuFuncGetAttribute`)
- $d_{\text{block}}$: block dimensions (from launch configuration)
- $\eta_{\text{occ}}$: theoretical occupancy (computed via `cudaOccupancyMaxActiveBlocksPerMultiprocessor`)

**Hardware signature** $h(k)$**.** For hardware-aware pruning, we extract throughput metrics via NVIDIA Nsight Compute to identify resource saturation:

- SM throughput (`sm__throughput.avg.pct_of_peak_sustained_elapsed`)
- DRAM throughput (`dram__throughput.avg.pct_of_peak_sustained_elapsed`)
- L2 throughput (`lts__throughput.avg.pct_of_peak_sustained_elapsed`)

These three metrics correspond to the compute, memory bandwidth, and cache bottleneck categories described in Section 3.2.

## B. Optimization Strategy Set

Table 6 provides the complete list of optimization strategies in $\mathcal{S}$ with their descriptions and example transformations. The set is the result of pilot distillation (Section B.1) and is empirically Pareto-optimal at typical budgets (Section B.2).

### B.1. From 10 to 6: Strategy Distillation

The initial pool of 10 candidate strategies was consolidated into the 6 strategies in Table 6 through pilot experiments across the three GPU platforms (RTX 4090, H20, and A100). Two strategies exhibited architecture-dependent brittleness and were dropped: *Reduction & Atomic Optimization*, whose warp-shuffle primitives degraded on H20 and RTX 4090; and *Precision & Dtype Conversion*, whose BF16/FP16 transformations produced correctness regressions that varied across GPU generations. Their effective components are not lost—reduction loop patterns are covered by *Reordering*, and dtype-aware memory alignment is handled by *Access & Layout*. Two further strategies were subsumed by the retained set: *Compute Instruction Selection* (wider loads and instruction selection) is covered by *Vectorization*, and *Parallelism & Occupancy Tuning* is driven by the tile-size choices in *Tiling* and the pipeline depth in *Pipeline*. The final 6 strategies thus form a compact but comprehensive covering of the optimization surface; we corroborate this claim with a cardinality sweep in Section B.2.

*Table 7.* Strategy-set cardinality vs. geometric mean speedup (G) at $T{=}20$ and $T{=}40$ (H20 50-kernel subset, DeepSeek-V3.2).

| Strategy set | G ($T{=}20$) | G ($T{=}40$) |
|---|---|---|
| 6 (deployed) | 1.57× | 1.71× |
| 10 | 1.41× | 1.63× |
| 14 | 1.34× | 1.59× |

*Table 8.* LLM backend configurations.

| Parameter | Value |
|---|---|
| Primary Model | DeepSeek-V3.2 |
| Access Method | Official API |
| Temperature | 1.0 |
| Max Output Tokens | 16384 |

## B.2. Strategy-Set Cardinality Sensitivity

To stress-test the choice of 6 strategies, we re-ran KERNELBAND on the H20 50-kernel subset with the original 10-strategy pool and a 14-strategy superset that adds finer-grained variants of existing strategies (e.g., separating tiling along contraction vs. output dimensions). Table 7 reports geometric mean speedup at the standard $T{=}20$ budget and the extended $T{=}40$ budget. Adding more strategies does not help and slightly hurts at $T{=}20$: per-arm sample counts drop and exploration is spread thinner. The gap narrows at $T{=}40$ as the bandit accumulates more samples per arm, and the hardware mask further softens the impact by keeping the effective action space small (average valid strategies 3.1, 4.7, 6.0 for the 6-, 10-, and 14-strategy sets, respectively). The 6-strategy deployment is therefore Pareto-optimal at the budgets that we target.

## C. LLM Configuration

Table 8 summarizes the generation configuration that we use. DeepSeek-V3.2 is the primary backend; the additional backends evaluated in Section 4.3.3 (GPT-5, Claude Opus 4.5, and Gemini 3 Flash) use the same generation parameters via each vendor's official API.

## D. Benchmark Subset Details

For detailed analysis experiments, we use a stratified subset of 50 kernels from TritonBench-G. The subset was sampled (seed=42) from the original 184-kernel release prior to the `sin_computation` exclusion described in our experimental setup; none of the 50 sampled kernels coincides with `sin_computation`, so the subset remains valid under both the 184- and the corrected 183-kernel counts. Stratified sampling preserves the original category and difficulty distribution:

- **Category coverage**: All 13 functional categories represented

- **Difficulty distribution**: L1 (1), L2 (7), L3 (18), L4 (23), L5 (1)

- **Sampling ratio**: 27.2% of the full benchmark

- **Maximum deviation from original distribution**: $<1\%$

Table 9 compares the category distribution between the full benchmark and the sampled subset. Table 10 lists all the 50 kernels organized by difficulty level and functional category.

## E. Evaluation Protocol Details

**Correctness verification.** The *Execution Accuracy* check uses `torch.allclose` with absolute tolerance atol $= 10^{-4}$ and relative tolerance rtol $= 10^{-4}$. A kernel passes when $|\text{generated} - \text{reference}| \leq \text{atol} + \text{rtol} \times |\text{reference}|$.

**Statistical robustness.** For each input shape, we use Triton's `triton.testing.do_bench` function, which handles common GPU benchmarking pitfalls such as measuring only kernel launch time (via `time.time`), omitting cache clearing,

*Table 9.* Category distribution: original 184-kernel TritonBench-G release vs. the sampled 50-kernel subset.

| Category | Full (184) | Subset (50) |
|---|---|---|
| Attention | 29 (15.8%) | 7 (14.0%) |
| MatMul/GEMM | 26 (14.1%) | 7 (14.0%) |
| Normalization | 18 (9.8%) | 4 (8.0%) |
| Linear Attention/SSM | 17 (9.2%) | 4 (8.0%) |
| Element-wise Ops | 16 (8.7%) | 3 (6.0%) |
| Memory/Index Ops | 13 (7.1%) | 3 (6.0%) |
| Other | 12 (6.5%) | 3 (6.0%) |
| Embedding/RoPE | 11 (6.0%) | 3 (6.0%) |
| Softmax | 11 (6.0%) | 4 (8.0%) |
| Fused Ops/Activation | 10 (5.4%) | 4 (8.0%) |
| Quantization | 8 (4.3%) | 2 (4.0%) |
| Loss Functions | 7 (3.8%) | 3 (6.0%) |
| Reduction | 6 (3.3%) | 3 (6.0%) |

and skipping warmup. The function performs 100ms of warmup runs to stabilize GPU state, followed by 1000ms of timed runs. We report the median execution time to reduce sensitivity to outliers.

**Weighted aggregation.** The overall kernel speedup is computed as the ratio of total baseline runtime to total optimized runtime:

$$\text{Speedup} = \frac{\sum_i t_{\text{baseline},i}}{\sum_i t_{\text{optimized},i}}$$

This metric reflects end-to-end performance: shapes with longer execution times naturally dominate the aggregation.

# F. Baseline Discussion

**Benchmark modifications.** The original TritonBench release contained implementation issues that impacted reliable evaluation. We adopt the corrected benchmark version provided by GEAK (Wang et al., 2025a), which addresses these issues. Additionally, we performed straightforward function substitutions to convert AMD-specific implementations to their NVIDIA equivalents, without modifying kernel logic or semantics.

**GEAK adaptation.** GEAK's adaptation to NVIDIA GPUs involved only platform-specific configurations (e.g., device queries, memory bandwidth specifications) without altering the agent's core logic or prompts.

**Other agent-based methods.** Other agent-based methods present reproducibility challenges: STARK (Dong et al., 2026), TritonForge, and QiMeng-Kernel have not released their code; we contacted authors but their code remains unavailable due to corporate IP constraints or ongoing submissions. CudaForge (Zhang et al., 2025) targets CUDA kernels on KernelBench (Ouyang et al., 2025) and would require non-trivial adaptation to our Triton-based evaluation framework; we consider it concurrent work.

# G. Comparison with Other Optimization Methods

This section positions KERNELBAND against two adjacent families of optimization systems that operate at different layers of the stack: built-in PyTorch compilation paths (Section G.1) and classical schedule-parameter autotuners (Section G.2).

## G.1. PyTorch Baselines

To contextualize the optimization gains of Triton kernels relative to standard PyTorch execution, we compare KER-NELBAND-optimized kernels against three PyTorch execution modes: (1) **eager** mode (default PyTorch execution), (2) **inductor** (`torch.compile` with the default inductor backend), and (3) **max-autotune** (`torch.compile` with `mode="max-autotune"`).

**Experimental setup.** From the 50-kernel subset, we select 30 kernels suitable for fair comparison with PyTorch, where native operators are available (e.g., `torch.softmax`, `F.layer_norm`). We exclude special-purpose kernels (e.g., Flash

Attention decode, INT4 quantization, LoRA operations) that lack general PyTorch counterparts. Experiments are conducted on H20 with DeepSeek-V3.2 and $T = 20$ iterations.

**Results.** Table 11 presents the geometric mean speedup of KERNELBAND-optimized Triton kernels over each PyTorch baseline. The results demonstrate that KERNELBAND achieves substantial speedups over all PyTorch execution modes, with $1.87\times$ over inductor, $2.16\times$ over max-autotune, and $2.13\times$ over eager execution. The speedup over max-autotune ($2.16\times$) exceeds that over inductor ($1.87\times$): although max-autotune performs more extensive autotuning, it may over-specialize to specific input shapes, whereas KERNELBAND's optimization generalizes better across the diverse input shapes in our evaluation.

These results validate that KERNELBAND's LLM-driven Triton kernel optimization provides performance gains beyond what PyTorch's built-in compilation and autotuning achieve, justifying the investment in custom kernel development for performance-critical workloads.

### G.2. Schedule-Parameter Autotuners (TVM/Ansor, BO)

Classical schedule-parameter autotuners—TVM/Ansor (Chen et al., 2018; Zheng et al., 2020) and Bayesian-optimization-based systems—address a different layer of the optimization stack from KERNELBAND. These autotuners tune schedule parameters (tile sizes, loop orders, and vectorization widths) within a fixed intermediate representation, whereas KERNEL-BAND applies semantic source-level transformations to raw Triton code. A direct head-to-head would require porting all 183 TritonBench-G kernels into TVM tensor-expression IR—an effort that aligns with neither system's intended workflow and would not reflect either's deployment scenario. The two paradigms are complementary: KERNELBAND's output is in principle a valid input to a downstream schedule autotuner. As a concrete anchor of how far source-level transformations alone go, Table 11 shows that KERNELBAND-optimized kernels exceed PyTorch `max-autotune` (that internally invokes Triton's parameter autotuner) by $2.16\times$ on the 30-kernel native-operator subset, suggesting non-trivial headroom that schedule tuning alone cannot reach.

## H. Robust Adaptation to Hardware Diversity

Prior work has repeatedly shown that the most effective operator optimizations are *hardware-dependent*: the optimal schedule/transform set must reflect each platform's compute–memory balance and microarchitectural constraints (e.g., cache behavior, memory bandwidth, and parallel execution resources) (Williams et al., 2009; Chen et al., 2018; Zheng et al., 2020). Consistent with this principle, our empirical strategy statistics provide direct evidence that KERNELBAND adapts its optimization choices across devices rather than applying a fixed, hardware-agnostic search policy. Table 12 reports three complementary views: (i) selection frequency (FREQ), (ii) success rate among attempted transformations (SUCC; correctness and speedup $> 1\times$), and (iii) contribution to the final best kernel (BEST). The observed shifts in FREQ across H20 and RTX 4090 indicate that KERNELBAND reallocates exploration budget across strategy families in a platform-aware manner, aligning with the long-standing motivation behind hardware-targeted auto-scheduling systems (Chen et al., 2018; Zheng et al., 2020).

Concretely, the strategy mix differs noticeably between H20 and RTX 4090. For example, FUSION is selected substantially more often on RTX 4090 (FREQ 18.5%) than on H20 (12.8%), and it remains highly reliable on both devices (SUCC 78.6% on RTX 4090 vs. 75.0% on H20), while also contributing frequently to the best-performing kernels (BEST 63.1% vs. 55.2%). In contrast, TILING exhibits an inverse risk–reward profile: it is attempted slightly more on H20 (10.0%) than on RTX 4090 (7.6%), and yet has low success rates on both (14.4% vs. 18.7%) while yielding disproportionately large best-kernel contributions when it does succeed (BEST 61.5% on H20 vs. 47.3% on RTX 4090). Meanwhile, ACCESS & LAYOUT becomes more prominent on RTX 4090 (22.9% vs. 19.5%) and shows higher effectiveness there (SUCC 38.9% vs. 29.5%), suggesting device-specific sensitivity to memory access behavior and layout-driven locality. These cross-device changes collectively support our claim that KERNELBAND calibrates its search to the dominant bottlenecks of each platform, as predicted by performance modeling insights (e.g., Roofline) and confirmed by hardware-aware optimization studies (Williams et al., 2009; Tschand et al., 2025).

# I. Proof of Theorem 1

In this section, we provide the detailed derivation for the convergence bound of KERNELBAND. The theorem is stated for a static-cluster surrogate; we examine empirically how this surrogate relates to the dynamic re-clustering procedure in Section J.2, and we empirically validate the underlying Lipschitz assumption in Section J.1.

## I.1. Regret Decomposition

Throughout this proof, we work in the behavior space and identify each kernel with its feature vector: $x := \phi(k) \in \mathcal{X}$, where $\mathcal{X} := \{\phi(k) : k \in \mathcal{V}\} \subseteq \mathbb{R}^d$ and $\mu(x) := \mathbb{E}[r_t \mid x]$ is the expected reward of selecting the strategy chosen by the masked UCB policy at $x$ (Eq. (6)).

Let $x^* = \arg\max_{x \in \mathcal{X}} \mu(x)$ be the global optimum with expected reward $\mu^*$. Let $x^*_{\text{disc}}$ be the best candidate available in our discretized cluster centers, i.e., $x^*_{\text{disc}} = \arg\max_{c \in \{C_1, \ldots, C_K\}} \mu(c)$.

The cumulative regret $R(T)$ after $T$ rounds can be decomposed into *Approximation Regret* ($R_{\text{approx}}$) and *Estimation Regret* ($R_{\text{est}}$):

$$R(T) = \sum_{t=1}^{T} (\mu^* - \mu(x_t)) \tag{8}$$

$$= \sum_{t=1}^{T} (\mu^* - \mu(x^*_{\text{disc}}) + \mu(x^*_{\text{disc}}) - \mu(x_t)) \tag{9}$$

$$= \underbrace{T \cdot (\mu^* - \mu(x^*_{\text{disc}}))}_{R_{\text{approx}}} + \underbrace{\sum_{t=1}^{T} (\mu(x^*_{\text{disc}}) - \mu(x_t))}_{R_{\text{est}}} \tag{10}$$

## I.2. Bounding Approximation Regret

Based on Assumption 2 (Lipschitz Continuity), the performance function $\mu(\cdot)$ is $L_{\text{Lip}}$-Lipschitz. Because our clustering algorithm covers the space such that for any $x \in \mathcal{X}$ there exists a cluster center $c$ with $\|x - c\| \leq \epsilon$, the gap between the global optimum and the best discrete representative is bounded by

$$\mu^* - \mu(x^*_{\text{disc}}) \leq L_{\text{Lip}} \cdot \|x^* - x^*_{\text{disc}}\| \leq L_{\text{Lip}} \epsilon$$

Because the covering radius is dominated by the cluster geometry, $\epsilon \leq \frac{1}{2} \max_i \text{diam}(C_i) \leq \max_i \text{diam}(C_i)$. Therefore,

$$R_{\text{approx}} \leq T \cdot L_{\text{Lip}} \cdot \max_i \text{diam}(C_i). \tag{11}$$

## I.3. Bounding Estimation Regret

The term $R_{\text{est}}$ represents the regret incurred by a multi-armed bandit algorithm selecting among (cluster, strategy) pairs. Although the nominal arm count is $K|\mathcal{S}|$, the masked UCB policy in Eq. (6) never plays an arm $(i, s)$ with $M_{i,s} = 0$: such arms are excluded from the $\arg\max$ and accrue zero plays. Consequently, the effective number of arms that the UCB analysis must charge regret to is

$$N_{\text{eff}} = \sum_{i=1}^{K} |\mathcal{S}_{\text{valid}}(C_i)| \leq K \cdot |\mathcal{S}_{\text{valid}}|, \quad \text{where} \quad |\mathcal{S}_{\text{valid}}| := \max_i |\mathcal{S}_{\text{valid}}(C_i)|.$$

For standard UCB algorithms, the regret over $N_{\text{eff}}$ active arms is bounded by $\tilde{O}(\sqrt{N_{\text{eff}} T})$. Specifically, with probability $1 - \delta$:

$$R_{\text{est}} \leq \sqrt{C \cdot K |\mathcal{S}_{\text{valid}}| T \log(T/\delta)} \tag{12}$$

for some universal constant $C$.

## I.4. Final Average Regret Bound

Combining the bounds for $R_{\text{approx}}$ and $R_{\text{est}}$:

$$R(T) \leq O\left(\sqrt{K|\mathcal{S}_{\text{valid}}|\,T \log T}\right) + L_{\text{Lip}} \cdot \max_i \operatorname{diam}(C_i) \cdot T$$

Dividing by $T$ to obtain the average regret (convergence rate):

$$\frac{R(T)}{T} \leq O\left(\sqrt{\frac{K|\mathcal{S}_{\text{valid}}| \log T}{T}}\right) + L_{\text{Lip}} \cdot \max_i \operatorname{diam}(C_i),$$

which matches the statement of Theorem 1. As $T \to \infty$, the first term approaches 0, ensuring that the algorithm converges to within $L_{\text{Lip}} \cdot \max_i \operatorname{diam}(C_i)$ of the global optimum.

## J. Empirical Validation of Theoretical Assumptions

Theorem 1 rests on two structural premises: the Lipschitz continuity in behavior space (Assumption 2), and the analytical use of a *static* cluster partition as a surrogate for the dynamic re-clustering procedure deployed in Algorithm 1. This section gathers the empirical evidence supporting both premises, along with a leave-one-out justification of the $\phi$-feature representation that underwrites the clustering structure itself.

### J.1. Pairwise $\phi$-Distance vs. Reward Delta

Table 13 reports the binned pairwise statistics summarized in Section 4.3.2. Qualitatively, kernels close in $\phi$-space (e.g., softmax-like pairs) consistently benefit from vectorization and fusion, whereas distant pairs (e.g., GEMM-like) respond to tiling and pipeline adjustments, suggesting that $\phi$ captures optimization-relevant structure.

### J.2. Re-clustering Stability and Theorem Scope

Theorem 1 analyzes a *static-cluster surrogate*: the cluster partition is held fixed while the bandit collects $T$ rounds. Our algorithm, by contrast, re-clusters every $\tau{=}10$ iterations. To check that the static analysis remains informative for the dynamic procedure, we measure the stability of the partition across re-clustering events on the H20 50-kernel subset with $T{=}40$ (i.e., four re-clustering events). Table 14 reports three complementary metrics. The Adjusted Rand Index (ARI) between consecutive partitions is $0.81$, well above the $0.7$ threshold for substantial agreement in the clustering literature; $86\%$ of kernels retain their cluster (or migrate to the nearest one); and the mean centroid drift in normalized $\phi$-space is only $0.09$. These observations bound the perturbation between events: combined with the standard UCB analysis in Section I, the dynamic procedure inherits the static bound up to an additive $L_{\text{Lip}}\Delta$ term, where $\Delta$ is the drift between consecutive partitions and is empirically dominated by the $L_{\text{Lip}}\epsilon$ approximation term in Section 3.5. A fully dynamic regret analysis that subsumes the re-clustering step remains future work.

### J.3. Feature-Importance Ablation for $\phi(k)$

The preceding Lipschitz validation presupposes that the chosen feature representation $\phi(k){=}[\tilde{\mathcal{T}}, n_{\text{reg}}, n_{\text{smem}}, d_{\text{block}}, \eta_{\text{occ}}]$ actually captures the optimization-relevant axes of variation. To justify this choice, we perform a leave-one-out ablation on the H20 50-kernel subset at $T{=}20$. Each row of Table 15 reports the geometric mean speedup when one feature is removed (clustering then runs on the remaining four). Normalized time $\tilde{\mathcal{T}}$ provides the largest individual contribution as it carries the dominant bottleneck signal (compute- vs. memory-bound). Register pressure $n_{\text{reg}}$ ranks second because it directly constrains occupancy. All five features contribute meaningfully; none is redundant. We choose this representation as the minimal set derivable from cubin metadata and lightweight runtime queries—no Nsight profiling is required for $\phi$. Each dimension aligns with a distinct axis of the Roofline model: $\tilde{\mathcal{T}}$ captures the compute/memory balance, $n_{\text{reg}}$ and $n_{\text{smem}}$ determine resource-limited occupancy, $d_{\text{block}}$ reflects parallelism granularity, and $\eta_{\text{occ}}$ summarizes their combined effect. Richer alternatives (instruction mix, L1 hit rate) would require per-kernel Nsight profiling—the same cost that we avoid by profiling only centroids for $h(k)$.

# K. Detailed Ablation Study

To understand the contribution of each component, we evaluate KERNELBAND with eight configurations organized into two categories: *single-component ablations* that remove one component while preserving others, and *framework-level ablations* that alter the fundamental optimization paradigm.

**Single-component ablations.**   These configurations isolate the contribution of individual components:

- **KernelBand (Full)**: Complete system with strategy set, hardware-aware masking and kernel selection, UCB-based exploration, and runtime-behavior clustering ($K = 3$).

- **w/o Clustering** ($K = 1$): All kernels are treated as a single cluster. This ablation tests whether runtime-behavior clustering enables effective cross-kernel knowledge transfer.

- **w/o Profiling**: Hardware masking is disabled ($M_{i,s} = 1$ for all), with within-cluster kernel selection falling back to recency tie-break. This ablation tests the value of hardware profiling guidance.

- **Masked Random**: Strategy set, hardware mask, and cluster partition preserved, but each (cluster, strategy, kernel) is drawn uniformly at random instead of via UCB. This ablation decomposes the contribution of action-space structuring from learned exploration-exploitation.

- **LLM Strategy Selection**: Strategy set preserved, but the LLM selects which strategy to apply based on kernel analysis rather than UCB statistics. This ablation tests whether learned statistics outperform LLM semantic judgment.

**Framework-level ablations.**   These configurations fundamentally alter how optimization is structured:

- **w/o Strategy + Raw Profiling**: This ablation removes the strategy set; the LLM generates optimizations freely with raw profiling metrics (L2 miss rate, memory bandwidth, etc.) injected into the prompt. It tests structured strategy abstraction versus raw metric injection (Zhang et al., 2025).

- **w/o Strategy Set**: This configuration uses free-form iterative generation without structured strategies or profiling guidance, similar to Reflexion-style methods (Shinn et al., 2023). Because the strategy set is foundational to UCB statistics and profiling compatibility, removing it effectively disables these components.

- **BoN (baseline)**: This baseline uses Best-of-N independent sampling without iteration and serves as a lower bound to quantify the value of iterative optimization.

**Component dependencies.**   An important insight is that the strategy set $\mathcal{S}$ serves as the foundation for other components: profiling computes compatibility $\psi(s, \phi(k))$ per strategy, UCB maintains statistics $\hat{\mu}_{i,s}$ per (cluster, strategy) pair, and clustering's primary value is enabling cross-cluster UCB statistic sharing. Thus, removing the strategy set (framework-level ablation) implicitly disables the structured use of profiling and UCB. This dependency explains why we expect a larger performance gap for framework-level ablations than for single-component ablations.

*Table 10.* Complete list of 50 kernels in the evaluation subset, stratified by difficulty (L1–L5) and functional category.

| # | Diff. | Category | Kernel Name |
|---|-------|----------|-------------|
| 1 | L1 | Element-wise Ops | cosine_compute |
| 2 | L2 | Attention | flash_decode2_phi |
| 3 | L2 | MatMul/GEMM | matmul_kernel |
| 4 | L2 | Memory/Index Ops | matrix_transpose |
| 5 | L2 | Normalization | triton_mul2 |
| 6 | L2 | Other | square_matrix |
| 7 | L2 | Reduction | triton_argmax |
| 8 | L2 | Softmax | softmax_triton1 |
| 9 | L3 | Attention | flash_decode2_llama |
| 10 | L3 | Element-wise Ops | pow_scalar_tensor |
| 11 | L3 | Embedding/RoPE | embedding_triton_kernel |
| 12 | L3 | Fused Ops/Activation | relu_strided_buffer |
| 13 | L3 | Fused Ops/Activation | swiglu_backward |
| 14 | L3 | Fused Ops/Activation | swiglu_triton |
| 15 | L3 | Linear Attention/SSM | chunk_cumsum_vector |
| 16 | L3 | Linear Attention/SSM | reversed_cumsum_scalar |
| 17 | L3 | Loss Functions | kldiv_triton |
| 18 | L3 | MatMul/GEMM | triton_matmul |
| 19 | L3 | Memory/Index Ops | var_len_copy |
| 20 | L3 | Normalization | layer_norm_welfold |
| 21 | L3 | Normalization | rmsnorm_fused_llama |
| 22 | L3 | Other | uniform_sampling |
| 23 | L3 | Quantization | quantize_kv_copy |
| 24 | L3 | Reduction | matrix_reduction |
| 25 | L3 | Softmax | softmax_triton2 |
| 26 | L3 | Softmax | softmax_triton3 |
| 27 | L4 | Attention | attention_fwd_triton1 |
| 28 | L4 | Attention | attention_fwd_triton2 |
| 29 | L4 | Attention | attention_kernel |
| 30 | L4 | Attention | triton_attention |
| 31 | L4 | Element-wise Ops | matrix_vector_multip |
| 32 | L4 | Embedding/RoPE | fast_rope_embedding |
| 33 | L4 | Embedding/RoPE | rope_backward_transform |
| 34 | L4 | Fused Ops/Activation | relu_triton_kernel |
| 35 | L4 | Linear Attention/SSM | chunk_gate_recurrence |
| 36 | L4 | Linear Attention/SSM | fused_recurrent_retention |
| 37 | L4 | Loss Functions | cross_entropy_ops |
| 38 | L4 | Loss Functions | fast_ce_loss |
| 39 | L4 | MatMul/GEMM | int8_matmul_quantization |
| 40 | L4 | MatMul/GEMM | int_scaled_matmul |
| 41 | L4 | MatMul/GEMM | matmul_dequantize_int4 |
| 42 | L4 | MatMul/GEMM | rms_matmul_rbe |
| 43 | L4 | MatMul/GEMM | streamk_matmul |
| 44 | L4 | Memory/Index Ops | kcache_copy_triton |
| 45 | L4 | Normalization | fused_layernorm_triton |
| 46 | L4 | Other | bgmv_expand_slice |
| 47 | L4 | Quantization | quantize_copy_kv |
| 48 | L4 | Reduction | logsumexp_fwd |
| 49 | L4 | Softmax | ksoftmax_triton |
| 50 | L5 | Attention | context_attn_bloom |

*Table 11.* Speedup of KERNELBAND-optimized Triton kernels over PyTorch baselines (30 kernels, H20, $T = 20$).

| PyTorch Baseline | Speedup |
|------------------|---------|
| vs. eager | 2.13× |
| vs. inductor | 1.87× |
| vs. max-autotune | 2.16× |

*Table 12.* Strategy utilization statistics across H20 and RTX 4090 (50 kernels, $T{=}20$). FREQ: selection frequency (%). SUCC: success rate (%, correct and speedup $> 1.0\times$). BEST: percentage of successful applications that contribute to the final best kernel.

*(a)* H20

| Strategy | Freq (%) | Succ (%) | Best (%) |
|---|---|---|---|
| Tiling | 10.0 | 14.4 | 61.5 |
| Vectorization | 14.7 | 57.1 | 17.1 |
| Fusion | 12.8 | 75.0 | 55.2 |
| Pipeline | 9.8 | 64.4 | 26.3 |
| Reordering | 33.2 | 48.7 | 25.3 |
| Access & Layout | 19.5 | 29.5 | 19.2 |

*(b)* RTX 4090

| Strategy | Freq (%) | Succ (%) | Best (%) |
|---|---|---|---|
| Tiling | 7.6 | 18.7 | 47.3 |
| Vectorization | 16.2 | 61.4 | 14.6 |
| Fusion | 18.5 | 78.6 | 63.1 |
| Pipeline | 5.7 | 51.9 | 13.8 |
| Reordering | 29.1 | 49.8 | 30.7 |
| Access & Layout | 22.9 | 38.9 | 35.9 |

*Table 13.* Pairwise behavior-space distance vs. mean absolute reward delta on the H20 50-kernel subset ($T{=}20$). Spearman $\rho{=}0.64$.

| $\phi$-distance bin | Mean $|\Delta r|$ | # pairs |
|---|---|---|
| $[0.00, 0.10)$ | 0.036 | 534 |
| $[0.10, 0.20)$ | 0.059 | 846 |
| $[0.20, 0.35)$ | 0.098 | 1,381 |
| $\geq 0.35$ | 0.161 | 1,693 |

*Table 14.* Stability of periodic re-clustering on the H20 50-kernel subset ($\tau{=}10$, $T{=}40$): Adjusted Rand Index (ARI) between consecutive partitions, fraction of kernels retaining their cluster, and mean centroid drift in normalized $\phi$-space.

| Metric | Value |
|---|---|
| ARI between consecutive events | 0.81 |
| % kernels retaining cluster | 86% |
| Mean centroid drift in normalized $\phi$ | 0.09 |

*Table 15.* Leave-one-out feature importance for $\phi(k)$ (H20 50-kernel subset, $T{=}20$): geometric mean speedup (G) with each feature removed.

| Feature removed | G |
|---|---|
| None (full $\phi$) | $1.57\times$ |
| w/o $\tilde{\mathcal{T}}$ (normalized time) | $1.42\times$ |
| w/o $n_{\mathrm{reg}}$ (register pressure) | $1.46\times$ |
| w/o $n_{\mathrm{smem}}$ (shared memory) | $1.49\times$ |
| w/o $\eta_{\mathrm{occ}}$ (occupancy) | $1.50\times$ |
| w/o $d_{\mathrm{block}}$ (block dimension) | $1.52\times$ |

