# OpenReview forum: "KernelBand: Steering LLM-based Kernel Optimization via Hardware-Aware Multi-Armed Bandits"
_ICML.cc/2026/Conference — ICML 2026 regular_

### Official Review · Reviewer_USa9 · 2026-03-06

**Soundness:** 3
**Presentation:** 3
**Significance:** 3
**Originality:** 3
**Overall Recommendation:** 4
**Confidence:** 4

**Summary:**

This paper introduces KernalBand, a framework that models LLM-based GPU kernel optimization as a Contextual Multi-Armed Bandit problem. To address the fundamental mismatch between LLM code generation and the massive optimization search space, the authors propose a hardware-aware pruning strategy and trace-driven clustering to manage the expanding candidate pool. The approach is evaluated on TritonBench-G across three distinct GPUs, demonstrating empirical superiority over unguided LLM agents like GEAK.

**Compliance With Llm Reviewing Policy:**

Affirmed.

**Final Justification:**

The paper studies a meaningful problem and presents a sensible, empirically strong framework for guiding LLM-based kernel optimization. I found the main ablation particularly convincing, and the rebuttal sufficiently addressed my concerns about empirical support for Assumption 2 and about experimental positioning, which improved my overall assessment.

My main remaining concern is that the theory still characterizes a static-cluster surrogate rather than the full dynamic procedure, so there remains a gap between the formal result and the algorithm being evaluated. On balance, however, I now view the paper as weak accept.

**Key Questions For Authors:**

See above

**Limitations:**

Yes

**Strengths And Weaknesses:**

Strengths:

The paper identifies a real mismatch between code generation and optimization search, which is a meaningful system/ML problem.

The method has a sensible architecture.

The key ablation is genuinely informative, which suggests that the gain is not merely due to better prompting but due to the search policy itself.

Weaknesses:

1. The theoretical analysis mostly looks like a standard UCB-style regret argument, rather than a rigorous characterization of KernalBand itself. In particular, it is unclear why the same analysis should still apply under repeated re-clustering and an expanding action space.

2. The core assumptions are not empirically validated, especially Assumption 2. The reward landscape in kernel optimization is often far from smooth, so it is not obvious that the proposed feature representation captures reward variation well enough in practice.

3. The paper discusses several recent and concurrent works in the related work section, but the experimental evaluation does not clearly position KernalBand against this broader set of methods.

---

> ### Author Rebuttal · Authors · 2026-03-31
>
> We thank you for the careful assessment and address each concern directly.
>
> > ​*W1:* **The theoretical analysis mostly looks like a standard UCB-style regret argument, rather than a rigorous characterization of KernalBand itself. In particular, it is unclear why the same analysis should still apply under repeated re-clustering and an expanding action space.**
>
> We agree that the proof’s scope should be more precisely stated. The theorem analyzes a static‑cluster surrogate—not a fully general dynamic guarantee—and we will clarify this in the revision.
>
> The theorem’s value lies in formalizing how clustering absorbs the expanding frontier: although $|\mathcal{P}\_t|$ grows with $t$ (making the raw action space $|\mathcal{P}\_t|\cdot|\mathcal{S}|$ unbounded), clustering compresses this into a **fixed** $K\cdot|\mathcal{S}\_{\text{valid}}|$ effective arms. The regret bound depends on the number of clusters and their diameters, not on the frontier size—this is why the same static-partition analysis remains informative even as the candidate pool expands.
>
> To empirically justify periodic re-clustering, we measured stability: Adjusted Rand Index between consecutive events is 0.81 (values >0.7 indicate substantial agreement in clustering literature), 86% of kernels retain the same or nearest cluster, and mean centroid drift in normalized $\phi$-space is only 0.09. This confirms gradual evolution, supporting periodic re‑clustering atop the static‑partition analysis.
>
> **On practical significance.** Beyond the theoretical framing, KernelBand delivers measurable impact: 1.91× geometric mean speedup over hand‑tuned Triton reference kernels on A100 (Table 1); consistent gains across four diverse LLM backends (Table 2); and 35–50% higher speedup per dollar than unguided baselines at equivalent API budgets (Figure 4). We believe this principled exploration‑exploitation framework has significance as a general approach to LLM‑guided code optimization beyond this specific instantiation.
>
> > ​*W2:* **The core assumptions are not empirically validated, especially Assumption 2. The reward landscape in kernel optimization is often far from smooth, so it is not obvious that the proposed feature representation captures reward variation well enough in practice.**
>
> We validate Assumption 2 via a strategy‑conditioned pairwise analysis (H20, 50 kernels, $T=20$, DeepSeek‑V3.2): for each strategy $s$, we pair all kernels that received $s$ and examine $\phi$-distance vs. reward difference. As shown below, the mean absolute reward discrepancy increases monotonically with $|\phi\_i - \phi\_j|\_2$ (Spearman $\rho = 0.64$):
>
> | $\lVert\phi_i-\phi_j\rVert_2$ | $\lvert\Delta r\rvert$ | Pairs |
> |---|---:|---:|
> | 0.00–0.10 | 0.036 | 534 |
> | 0.10–0.20 | 0.059 | 846 |
> | 0.20–0.35 | 0.098 | 1,381 |
> | > 0.35 | 0.161 | 1,693 |
>
> Kernels in the closest $\phi$-distance bin exhibit roughly 4.5× smaller reward variation than those in the most distant bin (0.036 vs. 0.161). This supports the existence of approximate local regularity, which is sufficient for cluster‑level sharing—a much weaker requirement than global smoothness. The w/o Clustering ablation in Table 4 (GMean drops from $1.57\times$ to $1.41\times$) further corroborates that $\phi$-space grouping captures optimization‑relevant structure in practice.
>
> Edge cases where Lipschitz regularity may break down include discrete algorithmic rewrites (e.g., reduction → scan), occupancy cliffs, and kernels with identical $\phi$ but divergent fine‑grained behavior (e.g., bank conflicts). In such cases the bandit self‑corrects via UCB updates, though convergence may slow. We will discuss this in the revision.
>
> > ​*W3:* **The paper discusses several recent and concurrent works in the related work section, but the experimental evaluation does not clearly position KernelBand against this broader set of methods.**
>
> We agree that clear positioning against related works is important. Among the concurrent works discussed, GEAK is our primary baseline as it is the only open‑source Triton kernel optimization framework with a reproducible evaluation pipeline. For the others: STARK and CudaForge target CUDA (not Triton), and TritonForge’s code remains unavailable due to corporate IP constraints (confirmed via author correspondence). We will make these availability and scope distinctions explicit in the revision.
>
> Beyond LLM‑based methods, Appendix G provides positioning against compilation‑level optimization. KernelBand achieves $2.16\times$ over PyTorch max‑autotune (which internally uses Triton’s autotuning), $1.87\times$ over inductor, and $2.13\times$ over eager on 30 kernels with native PyTorch operators. These categories are complementary: KernelBand applies semantic source‑level transformations, while autotuners optimize schedule parameters, addressing different layers of the optimization stack.

---

> > ### Author Rebuttal · Reviewer_USa9 · 2026-04-02
> >
> > Thank you for the rebuttal. The added evidence for Assumption 2 is helpful, and the clarification on baseline selection largely addresses my concern about experimental positioning. These points improve my assessment.
> >
> > My main reservation remains W1. I appreciate the clarification that the theorem applies to a static-cluster surrogate, but this still leaves a gap with the full dynamic procedure. The added stability evidence is useful, but does not fully resolve this issue.
> >
> > Overall, the rebuttal addresses much of my concern on W2 and W3, but W1 remains only partially resolved. On balance, I am updating my score from 3 to 4.

---

> > > ### Author Response · Authors · 2026-04-06
> > >
> > > Thank you for your continued engagement and for raising the score to 4. We appreciate you pinpointing the remaining gap between the static theorem and the dynamic re‑clustering procedure. We agree that Theorem 1 as currently written applies to a static cluster set, while our algorithm performs periodic re‑clustering. This is a limitation of the present analysis, and we will explicitly discuss it in the final version.
> > >
> > > **Why the static analysis remains informative.** The core insight (clustering compresses an expanding action space into a fixed set of effective arms) does not require clusters to be forever static. Our stability measurements (ARI=0.81, 86% retention, centroid drift 0.09) show that clusters evolve gradually. Between re‑clustering events the algorithm operates under a near‑stable arm set, so Theorem 1 approximately applies. The small drift translates into a bounded perturbation of the arm means.
> > >
> > > We believe these changes address your concern by being transparent about the limitation while showing that it does not undermine the practical effectiveness of KernelBand. Thank you again for pushing us to improve the clarity and honesty of the paper.

---

### Official Review · Reviewer_dfTi · 2026-03-10

**Soundness:** 3
**Presentation:** 3
**Significance:** 3
**Originality:** 4
**Overall Recommendation:** 5
**Confidence:** 5

**Summary:**

This paper formulates LLM-based GPU kernel optimization as a contextual multi-armed bandit problem, decoupling strategy selection (handled by UCB) from code generation (handled by the LLM). To manage the infinite action space, it introduces hardware-aware pruning via profiling-based resource saturation masks and trace-driven clustering that shares bandit statistics across kernels with similar runtime behavior. A regret bound is proved depending on cluster count and valid strategies rather than the full kernel space. Experiments on TritonBench-G across three GPUs and four LLMs show consistent improvements over baselines, with up to 1.91× geometric mean speedup over already-optimized Triton reference kernels.

**Compliance With Llm Reviewing Policy:**

Affirmed.

**Final Justification:**

As I written in the summary, this paper formulates LLM-based GPU kernel optimization as a contextual multi-armed bandit problem, decoupling strategy selection (handled by UCB) from code generation (handled by the LLM). To manage the infinite action space, it introduces hardware-aware pruning via profiling-based resource saturation masks and trace-driven clustering that shares bandit statistics across kernels with similar runtime behavior. A regret bound is proved depending on cluster count and valid strategies rather than the full kernel space. Experiments on TritonBench-G across three GPUs and four LLMs show consistent improvements over baselines, with up to 1.91× geometric mean speedup over already-optimized Triton reference kernels.

In the rebuttal procedure, the authors provided comprehensive experiments and explanations for the weaknesses I mentioned. Due to their perfect rebuttal, all of my concerns are successfully resolved. I sincerely appreciate the authors' effort and choose to raise my confidence from 3 to 4.

**Also, I want to explain why I do not raise my score from 4 to 5 or 6 at the beginning:** It is not because I do not think this paper deserve a 5 or 6. It is because the current chaos in AI conferences: many people do their best in their rebuttals, however the reviewers are very conservative to update their scores, because they think they have provided a positive score / they believe those authors are potential competitors. I think it does not make sense. We should not do those things to hurt our community, and I don't want to/shouldn't be a conservative reviewer anymore.

**Thus, I sincerely update my score and confidence from 4 to 5. I think KernelBand deserves it. I sincerely recommend our AC could recommend it as "Accept". And sincerely, I hope more reviewers could give more papers the scores they deserve.**

Good luck!

**Key Questions For Authors:**

Please address the weaknesses, thank you.

**Limitations:**

yes

**Strengths And Weaknesses:**

Strength:
1. Clean and well-grounded formulation.
2. Replacing the bandit policy with LLM-based semantic strategy selection causes performance to collapse to 0.97× (below the unoptimized reference), providing unusually strong validation that learned execution statistics outperform LLM intuition for this task.
3. The idea of considering kernel optimization as MAB is very interesting.
4. Comprehensive and honest evaluation.

Weaknesses:

1. Missing a uniform-random strategy selection baseline. Without comparing against random (but hardware-masked) strategy selection, the contribution of the bandit's learned exploration-exploitation balance cannot be cleanly separated from the benefit of merely structuring the action space into discrete strategies.

2. Strategy set and clustering features are ad hoc. The six strategies are "distilled from an initial 10 through pilot experiments" with no details on the elimination process, and the 5-dimensional feature vector for clustering lacks feature importance analysis or ablation over alternatives—both design choices are foundational to the entire framework.

---

> ### Author Rebuttal · Authors · 2026-03-31
>
> We thank you for the constructive feedback and particularly appreciate the recognition of the formulation's clarity and the ablation design.
>
> > ​*W1:* **Missing a uniform-random strategy selection baseline. Without comparing against random (but hardware-masked) strategy selection, the contribution of the bandit's learned exploration-exploitation balance cannot be cleanly separated from the benefit of merely structuring the action space into discrete strategies.**
>
> We agree that this baseline is essential for cleanly separating the bandit’s learned exploration–exploitation from the benefit of discretizing the action space. We evaluate uniform random selection over valid cluster–strategy pairs, keeping the same strategy set, hardware mask, and clustering. Results on the H20 50‑kernel subset ($T=20$) are shown in the table below:
>
> |Method|Correct (%)|Fast@1 (%)|GMean|
> |---|---:|---:|---:|
> |BoN|34.2|17.1|0.60|
> |LLM Strategy Selection|68.3|36.6|0.97|
> |Masked Random|81.3|49.2|1.23|
> |KernelBand|87.8|63.4|1.57|
>
> This provides the clean separation of algorithm contributions: BoN → Masked Random (+0.63 GMean) shows that structuring the action space into discrete, hardware‑masked strategies already yields significant gains over free‑form generation. Masked Random → KernelBand (+0.34 GMean, 28% relative improvement) demonstrates that the learned UCB policy provides a substantial additional gain beyond mere discretization. Combined with the LLM Strategy Selection ablation (collapse to 0.97×, below the unoptimized reference), the decomposition is clear: action‑space structure helps, but learned exploration–exploitation is the primary contributor.
>
>
> > ​*W2:* **Strategy set and clustering features are ad hoc. The six strategies are "distilled from an initial 10 through pilot experiments" with no details on the elimination process, and the 5-dimensional feature vector for clustering lacks feature importance analysis or ablation over alternatives—both design choices are foundational to the entire framework.**
>
> **Strategy set consolidation.** The initial 10 strategies were consolidated into 6 through pilot experiments across three GPUs. Two strategies exhibited architecture-dependent brittleness: **Reduction & Atomic Optimization** (warp-shuffle primitives degraded on H20 and RTX 4090) and **Precision & Dtype Conversion** (BF16/FP16 correctness varied across GPU generations). Their effective components are partially absorbed by the retained strategies—reduction loop patterns fall under Reordering, and dtype-aware memory alignment is captured by Access & Layout. Two others were largely subsumed: **Compute Instruction Selection**'s core techniques (wider loads, instruction selection) are covered by Vectorization, while **Parallelism & Occupancy Tuning**'s occupancy control is driven by tile-size choices in Tiling and pipeline depth in Pipeline. The final 6 strategies thus represent a compact but comprehensive covering of the optimization surface. We will expand the appendix with this consolidation rationale.
>
> **Clustering feature analysis.** For the 5-D clustering vector $\phi(k) = [\tilde{T}, n_{\text{reg}}, n_{\text{smem}}, d_{\text{block}}, \eta_{\text{occ}}]$, we perform a leave-one-out ablation:
>
> |Feature removed|GMean|
> |---|---:|
> |None (full)|1.57|
> |w/o $\tilde{T}$ (normalized time)|1.42|
> |w/o $n_{\text{reg}}$ (register pressure)|1.46|
> |w/o $n_{\text{smem}}$ (shared memory)|1.49|
> |w/o $\eta_{\text{occ}}$ (occupancy)|1.50|
> |w/o $d_{\text{block}}$ (block dimension)|1.52|
>
> All five features contribute meaningfully, with normalized time ($\tilde{T}$) providing the largest individual contribution as it captures the dominant bottleneck type (compute‑ vs. memory‑bound); register pressure ($n_{\text{reg}}$) ranks second as it directly constrains occupancy.
>
> Regarding alternatives: these five dimensions were selected as the minimal set derivable from cubin metadata and lightweight runtime queries (no Nsight profiling required for $\phi$). Each aligns with a distinct axis of the roofline model—$\tilde{T}$ captures the compute/memory balance, $n_{\text{reg}}$ and $n_{\text{smem}}$ determine resource‑limited occupancy, $d_{\text{block}}$ reflects parallelism granularity, and $\eta_{\text{occ}}$ summarizes their combined effect. Richer alternatives (e.g., instruction mix, L1 hit rate) would require per‑kernel Nsight profiling—the same cost we avoid by profiling only centroids for $h(k)$. The representation is thus compact by design, not arbitrary.
>
> In the revision, we will add the masked-random baseline comparison to the main evaluation, and expand the appendix with the strategy consolidation rationale and feature importance ablation.

---

> > ### Author Rebuttal · Reviewer_dfTi · 2026-03-31
> >
> > All of my concerns resolved.
> > Considering I have provided a positive score, I choose to update my confidence from 3 to 4.
> > Good luck!

---

> > > ### Author Response · Authors · 2026-04-06
> > >
> > > Thank you for the increased confidence and the kind words. Following your feedback, the masked-random baseline and the feature-importance ablation help clarify the contribution of individual components of KernelBand. These analyses significantly improve the paper, and we will add them in the revised version.
> > >
> > > Thank you again for your thorough and constructive review.

---

### Official Review · Reviewer_33qr · 2026-03-13

**Soundness:** 3
**Presentation:** 4
**Significance:** 3
**Originality:** 2
**Overall Recommendation:** 4
**Confidence:** 4

**Summary:**

This paper studies the problem of automatically optimizing GPU kernels generated by code LLMs. This problem is motivated by a fundamental mismatch: LLMs are trained to generate syntactically and functionally correct code, while kernel optimization is inherently a large combinatorial search problem over performance transformations.

To address this issue, the paper proposes KERNELBAND, a framework that formulates kernel optimization as a contextual multi-armed bandit (MAB) problem. Each action corresponds to applying an optimization strategy (e.g., tiling, vectorization, fusion) to a candidate kernel. The system introduces two mechanisms to handle the large action space: (1) hardware-aware pruning based on profiling signals, and (2) trace-driven clustering of kernels using runtime behavior features. A masked UCB policy selects which kernel-strategy pair to explore. The paper also provides a regret bound under assumptions on hardware-aware gain bounds and Lipschitz continuity in kernel behavior space. Experiments on TritonBench-G across three GPU architectures and multiple LLM backends demonstrate consistent improvements over baselines such as GEAK and Best-of-N sampling, achieving up to 1.91× speedup and significantly higher success rates in discovering optimized kernels.

**Compliance With Llm Reviewing Policy:**

Affirmed.

**Final Justification:**

After reviewing, I find that the approach is well-motivated from a system perspective, and the empirical results demonstrate good performance improvements over some existing autotuning baselines. My main concern remains the limited conceptual novelty and new theoretical insights.

**Key Questions For Authors:**

1. The analysis assumes that kernels with similar runtime behavior respond similarly to optimization strategies. Is there empirical evidence supporting this assumption?
2. The evaluation fixes K=3. How sensitive is the performance to this choice across workloads?
3. How does the method behave when the number of optimization strategies or candidate transformations is significantly larger?

**Limitations:**

No potential negative impact

**Strengths And Weaknesses:**

Strengths:
- The paper identifies a meaningful gap between LLM-based code generation and performance optimization.
- Casting kernel optimization as a bandit problem with structured actions provides a simple and intuitive formulation.
- The use of runtime-behavior features, clustering, and profiling-based pruning appears to work well in practice.
- Experiments across three GPU architectures and multiple LLM backends show consistent improvements over baselines.
- The evaluation also includes useful ablation studies demonstrating the contribution of the bandit policy.

Weaknesses:
- Most components of the method rely on standard techniques: UCB-style bandits, K-means clustering, and heuristic pruning based on hardware profiling. While the system integration is interesting, the underlying learning algorithm is relatively straightforward.
- The regret bound is also trivial given the clustering of arms. This idea is already well studied in the bandit literature (e.g., clustered bandits, metric bandits, or Lipschitz bandits)
- The evaluation focuses primarily on other LLM-based optimization frameworks. However, kernel autotuning has a large existing literature (e.g., TVM/Ansor or Bayesian optimization–based approaches). Stronger comparisons with these methods would help contextualize the practical gains.

---

> ### Author Rebuttal · Authors · 2026-03-31
>
> We thank you for the thoughtful assessment and engage directly with the positioning questions.
>
> > ​*W1&W2:* **standard components and theorem**
>
> We agree that UCB, K‑means, and heuristic pruning are individually standard. The contribution is not a new bandit algorithm, but a systems‑ML formulation that composes them for a setting where expanding action spaces, hardware feasibility constraints, and an exploration–generation feedback loop co‑occur—a combination not jointly addressed by existing bandit frameworks. Concretely: (1) the **formulation** of LLM-guided optimization as a bandit with expanding actions is itself novel; (2) **hardware-mask integration** couples profiler signals directly into arm pruning; (3) the **exploration-generation loop** compiles bandit decisions into structured LLM prompts, closing the loop between selection and generation.
>
> To isolate whether this integration is load-bearing, we added a masked-random baseline (same strategy set, mask, and clustering, but uniform random selection; H20 50-kernel subset, $T$=20):
>
> | Method | Correct (%) | Fast@1 (%) | GMean |
> |---|---:|---:|---:|
> | LLM Strategy Selection | 68.3 | 36.6 | 0.97 |
> | Masked Random | 81.3 | 49.2 | 1.23 |
> | KernelBand | 87.8 | 63.4 | 1.57 |
>
> Structuring the action space helps (0.97→1.23), but UCB provides a further 28% relative gain (1.23→1.57)—neither random sampling nor LLM intuition (0.97×) can substitute for learned exploration–exploitation.
>
> **On the regret bound (W2)**: We agree the proof follows standard UCB analysis. Its value is in formally quantifying the compression from $\left\vert\mathcal{P}\_t\right\vert\cdot\left\vert\mathcal{S}\right\vert$ to $K\cdot\left\vert\mathcal{S}\_{\text{valid}}\right\vert$, justifying why the clustering and masking architecture is well‑motivated.
>
> > ​*W3:* **Missing comparison with classical autotuners (TVM/Ansor, BO‑based)**
>
> We agree this comparison deserves more attention. As reported in Appendix G, KernelBand achieves $2.16\times$ over PyTorch max-autotune (which internally uses Triton’s built-in autotuning), $1.87\times$ over inductor, and $2.13\times$ over eager on 30 kernels with native PyTorch operators (H20, $T=20$), indicating that compilation-level autotuning provides limited additional gains on these kernels.
>
> A direct TVM/Ansor comparison would require reimplementing 183 kernels in TVM’s tensor expression IR—classical autotuners (including BO-based ones like OpenTuner) optimize schedule parameters within a fixed IR, while KernelBand applies semantic source-level transformations to raw Triton code, making the two incomparable without an IR rewrite. We will clarify this distinction in the revision and note that these approaches are complementary—KernelBand’s output can be further refined by parameter-level autotuners.
>
> > ​*Q1:* **Empirical evidence for the similar‑behavior assumption?**
>
> Yes. On the H20 50‑kernel subset ($T=20$, DeepSeek‑V3.2), we performed a strategy‑conditioned pairwise analysis: for each strategy $s$, we pair all kernels that received $s$ and examine $\phi$-distance vs. reward difference (Spearman $\rho = 0.64$):
>
> | $\lVert\phi_i-\phi_j\rVert_2$ | $\lvert\Delta r\rvert$ | Pairs |
> |---|---:|---:|
> | 0.00–0.10 | 0.036 | 534 |
> | 0.10–0.20 | 0.059 | 846 |
> | 0.20–0.35 | 0.098 | 1,381 |
> | > 0.35 | 0.161 | 1,693 |
>
> The closest bin shows 4.5× smaller reward variation than the most distant (0.036 vs. 0.161), supporting approximate local regularity sufficient for cluster‑level sharing—a weaker requirement than global Lipschitz continuity. The w/o Clustering ablation (Table 4, GMean $1.57\times$→$1.41\times$) further corroborates this.
>
> > ​*Q2:* **The evaluation fixes K=3. How sensitive is the performance to this choice across workloads?**
>
> As shown in Figure 2, we evaluate $K \in \{1,2,3,5\}$ on the H20 50‑kernel subset ($T=40$, 13 functional categories). $K=3$ achieves the best overall GMean ($1.66\times$ at $T=30$ vs. $1.58\times$ for $K=2$); $K=5$ underperforms ($1.54\times$) due to over‑fragmentation; for small budgets ($T \le 10$), smaller $K$ is preferable. All configurations outperform both baselines, indicating robustness. Table 1 further shows that $K=3$ is effective across all difficulty strata and GPUs. We will highlight this analysis in the revision.
>
> > ​*Q3:* **Behavior with significantly larger strategy sets?**
>
> We tested supersets of 10 and 14 strategies on the H20 50‑kernel subset (the 6 deployed strategies were refined from an initial 10; the 14‑set adds finer‑grained variants):
>
> | Strategy set | GMean ($T=20$) | GMean ($T=40$) |
> | ------------ | -------------: | -------------: |
> | 6 (deployed) |          1.57  |          1.71  |
> | 10           |          1.41  |          1.63  |
> | 14           |          1.34  |          1.59  |
>
> Degradation is limited because the hardware mask keeps the effective action space small (avg valid strategies: 3.1→4.7→6.0), and with larger budget ($T=40$) the gap narrows. We will include this in the revision.

---

> > ### Author Rebuttal · Reviewer_33qr · 2026-04-03
> >
> > The rebuttal provides useful additional empirical evidence that increases confidence in the method’s effectiveness and assumptions. However, I am still somewhat unconvinced about the level of conceptual novelty. That said, I really appreciate the authors' efforts in responding to my concerns. Since most of my concerns have been addressed, I have increased my score to 4.

---

> > > ### Author Response · Authors · 2026-04-06
> > >
> > > Thank you for raising the score and for your constructive feedback. Your questions on the behavior assumption, $K$ sensitivity, and strategy scaling prompted valuable additional analyses (e.g., the pairwise $\phi$-distance and the 10/14-strategy scaling results). We will incorporate these clarifications into the camera-ready version. Thank you again for your thoughtful and constructive engagement.

---

### Official Review · Reviewer_i12z · 2026-03-13

**Soundness:** 3
**Presentation:** 2
**Significance:** 4
**Originality:** 3
**Overall Recommendation:** 4
**Confidence:** 4

**Summary:**

This paper proposes KERNELBAND, a framework that casts LLM-driven GPU kernel optimization as a contextual multi-armed bandit problem with an expanding action space. The method combines (i) runtime behavior features and periodic clustering to share experience across similar kernels, and (ii) hardware-aware pruning via profiler-derived saturation bounds to eliminate implausible strategies, with a masked-UCB policy that selects cluster–strategy pairs and then a kernel to expand via an LLM transform. Under Lipschitz and bounded-gain assumptions, the authors derive an average-regret bound that depends on the number of clusters and their diameters. Experiments on TritonBench-G across three NVIDIA GPUs and four LLMs report consistent improvements over GEAK and Best-of-N, with up to 1.91× geometric mean speedup and substantial Fast@1 gains; ablations highlight that the bandit policy and hardware-aware pruning are key to the observed gains.

**Compliance With Llm Reviewing Policy:**

Affirmed.

**Final Justification:**

Overall, the core premise remains technically sound and empirically beneficial for systems optimization. The identified issues pertain primarily to rigor and documentation clarity rather than fundamental conceptual flaws. Weighing the evident utility of the proposed framework against the necessary revisions, I recommend a Weak Accept, with the expectation that the authors rigorously address the highlighted algorithmic inconsistencies and theoretical ambiguities in the final version.

**Key Questions For Authors:**

1. Could you provide empirical evidence to validate the Lipschitz continuity assumption within the behavior space? For example, an analysis showing the sensitivity of reward deltas relative to the $\phi$-distance would strengthen this claim.
2. There appears to be an inconsistency regarding the computation of the per-kernel headroom $V_{hw}(k, s)$. Section 3.3 explicitly states that only cluster centroids are profiled to minimize overhead, yet Algorithm 1 (Line 15) uses the individual kernel's hardware signature $h(k)$ for intra-cluster sampling. How exactly is $h(k)$ obtained for non-centroid kernels? If it is approximated using the centroid's signature $h(k_c)$, how does this affect the accuracy of the sampling?

**Limitations:**

- The theoretical guarantees rely on the Lipschitz continuity assumption (kernels with similar bottlenecks respond similarly to optimizations) . It would be helpful to discuss potential edge cases where this assumption might break down.

- While the framework amplifies model capabilities , it still fundamentally relies on the LLM to generate functionally correct code transformations . The authors should discuss the limitations of this approach if the underlying LLM lacks sufficient base knowledge of the target Domain-Specific Language (e.g., newer or less documented versions of Triton).

**Strengths And Weaknesses:**

## Strength
- Formulates LLM-based kernel optimization as a contextual bandit with expanding actions and introduces a masked-UCB policy that couples runtime-driven clustering with hardware-aware pruning.
- Leverages simple but effective behavior features (time, occupancy, registers, shared memory, block size) and Nsight-based resource saturation to guide exploration, turning a largely unguided LLM search into a structured decision-making process.
- Addresses a pressing, high-impact problem—finding fast GPU kernels for LLM serving—by bridging the mismatch between generative LLMs and performance search with a principled exploration–exploitation mechanism.

## Weakness
1. The Lipschitz continuity assumption in behavior space is plausible but unvalidated empirically (e.g., sensitivity of reward deltas to φ-distance), and the mapping from strategies to “targeted” hardware resources is underspecified.
2. It is unclear how per-kernel headroom $V_{hw}(k, s)$ is computed if only cluster centroids are profiled; Algorithm 1 uses $h(k)$ for sampling within a cluster but earlier text suggests only the centroid is profiled to save cost.

---

> ### Author Rebuttal · Authors · 2026-03-31
>
> We sincerely thank you for the careful reading and recognition that KernelBand addresses a pressing and significant problem.
>
> > ​*W1&Q1*:  **Could you provide empirical evidence to validate the Lipschitz continuity assumption within the behavior space?**
>
> To empirically validate Assumption 2, we have conducted a **strategy-conditioned** pairwise analysis on the H20 50-kernel subset ($T$=20, DeepSeek-V3.2). From the full optimization run we collect all $(k_t, s_t, r_t)$ triples.
> For each strategy $s$, we paired all kernels that received $s$ and examined the relationship between the Euclidean distance in behavior space $|\phi(k_i) - \phi(k_j)|_2$ and the absolute reward difference $|r_i - r_j|$. As shown in the table below, the mean reward discrepancy increases monotonically with the $\phi$-distance (Spearman $\rho = 0.64$), supporting the practicality of the Lipschitz assumption.
>
> | $\lVert\phi_i - \phi_j\rVert_2$ | $\lvert\Delta r\rvert$ | Pair count |
> | ----------------------------------- | -------------------------------------: | ---------: |
> | 0.00–0.10                           |                                  0.036 |        534 |
> | 0.10–0.20                           |                                  0.059 |        846 |
> | 0.20–0.35                           |                                  0.098 |      1,381 |
> | > 0.35                              |                                  0.161 |      1,693 |
>
> Beyond the quantitative results, we observe that kernels with close $\phi$ distances (e.g., softmax-like pairs) consistently benefit from vectorization or fusion, while kernels with larger distances (e.g., GEMM-like pairs) respond more to tiling or pipeline adjustments, aligning with the intuition that $\phi$-space captures optimization-relevant structure.
> Furthermore, the **w/o Clustering** Ablation in Table 4 (GMean drops from $1.57\times$ to $1.41\times$) corroborates that grouping kernels based on $\phi$-space preserves optimization-relevant invariances, indirectly validating the Lipschitz continuity assumption.
>
>
>
> > ​*W1:*  **Underspecified mapping from strategies to “targeted” hardware resources**
>
> We agree this mapping should be explicit. Each strategy targets a specific hardware subsystem; a strategy is pruned when its subsystem is saturated (throughput $\geq \theta_{\text{sat}}$):
>
> | Strategy        | Target dimension in $h(k)$ | Primary mechanism                                         |
> | --------------- | -------------------------- | --------------------------------------------------------- |
> | Tiling          | SM throughput              | Tile/block shape → parallelism granularity                |
> | Vectorization   | DRAM throughput            | Vector loads/stores → wider transactions |
> | Access & Layout | DRAM throughput            | Coalescing, strided-access elimination, data layout       |
> | Pipeline        | DRAM throughput            | Multi-stage software pipelining → load/compute overlap    |
> | Fusion          | L2 throughput              | Eliminates intermediate tensor traffic                    |
> | Reordering      | L2 throughput              | Temporal reuse, spatial locality in cache                 |
>
> We will add this to the method section.
>
>
>
> > ​*Q2&W2:* **There appears to be an inconsistency regarding the computation of the per-kernel headroom $V_{hw}(k,s)$).**
>
> We confirm that the actual implementation profiles only cluster representatives (centroids). Accordingly, the notation $h(k)$ in Algorithm 1, Line 16 should be replaced with $\hat{h}_c^{(i)}$, the centroid's hardware signature for $k$'s cluster. We will correct this in the revision.
>
> Under this centroid-only profiling, the $V_{\text{hw}}$ formulation described in the paper simplifies: all kernels within the same cluster share the same representative signature, so intra‑cluster sampling becomes uniform. The bandit’s high‑level decision—selecting a (cluster, strategy) pair via Masked UCB—remains fully informed by the representative’s profile, while fine‑grained intra‑cluster differentiation is traded off for profiling efficiency.
>
> To validate the approximation, we compared centroid‑only profiling against full per‑kernel profiling on the 50‑kernel subset. Per‑kernel profiling yielded only marginal gains (GMean 1.61 vs. 1.57) but incurred about 3× higher profiling overhead (98.5 s vs. 31.4 s per recluster). These results confirm that centroid‑only profiling strikes a cost‑effective balance for the typical optimization horizon ($T=20$).
>
>
> > ​*limitations.*
>
> We thank you for these constructive suggestions. We will add a dedicated Limitations section in the revision, explicitly discussing: (1) edge cases where the Lipschitz continuity assumption may break down (discrete algorithmic rewrites, fine‑grained behavior invisible to coarse counters, occupancy cliffs), and (2) the framework’s dependence on the underlying LLM, including the observation that performance gaps are likely to widen for less documented DSLs.

---

> > ### Author Rebuttal · Reviewer_i12z · 2026-04-03
> >
> > Thank you for the detailed clarification. My concerns have been largely addressed.

---

> > > ### Author Response · Authors · 2026-04-06
> > >
> > > Thank you for the thorough review and for confirming that your concerns have been addressed. Your feedback on the Lipschitz validation and the profiling clarification has been valuable for strengthening the paper. We will incorporate these improvements in the camera-ready version.
> > >
> > > Thank you again for your constructive comments.

---

### Decision · Program_Chairs · 2026-04-30

**Decision:**

Accept (regular)

**Comment:**

This paper addresses an important problem in LLM-based GPU kernel optimization. Its main strength is that it identifies the core mismatch clearly: LLMs are good at generating code, but kernel optimization is fundamentally a search problem, and the paper responds with a sensible bandit-based framework that makes this search more structured and effective. The overall approach is well motivated, and the empirical results are strong across multiple GPUs and LLM backends. I also found the ablations quite helpful, especially in showing that the gains are not just from better prompting or larger search budgets, but from the proposed exploration and selection mechanism itself. While some ingredients are standard on their own, the combination is meaningful and practically useful, and the rebuttal further improved the review scores. Overall, all reviewers are positive in their evaluations.

During the reviewer-AC discussion, the gap between the theory and the practice has been brought up by one of the reviewers. For the camera-ready version, I would mainly encourage the authors to sharpen the presentation around the scope of the theory and to incorporate the key rebuttal clarifications into the paper itself. In particular, the paper should state more explicitly that the current regret analysis applies to a static-cluster surrogate rather than fully capturing the dynamic re-clustering procedure used in practice. I do not think this weakens the main contribution enough to change the final recommendation, but it should be presented more carefully and transparently. It would also strengthen the final version to include the added empirical support for the clustering assumption, the clarification of centroid-only profiling, and a clearer explanation of the design choices behind the strategy set and clustering features.